# ANIMATE-X: UNIVERSAL CHARACTER IMAGE ANIMATION WITH ENHANCED MOTION REPRESENTATION

**Shuai Tan**[1*], **Biao Gong**[1†], **Xiang Wang**[2], **Shiwei Zhang**[2],
**Dandan Zheng**[1], **Ruobing Zheng**[1], **Kecheng Zheng**[1], **Jingdong Chen**[1], **Ming Yang**[1]

[1]Ant Group    [2]Alibaba Group

{tanshuai2001,a.biao.gong}@gmail.com,
{xiaolao.wx,zhangjin.zsw}@alibaba-inc.com,{yuandan.zdd,
zhengruobing.zrb,zhengkecheng.zkc,jingdongchen.cjd,m.yang}@antgroup.com

Project Page: https://lucaria-academy.github.io/Animate-X/

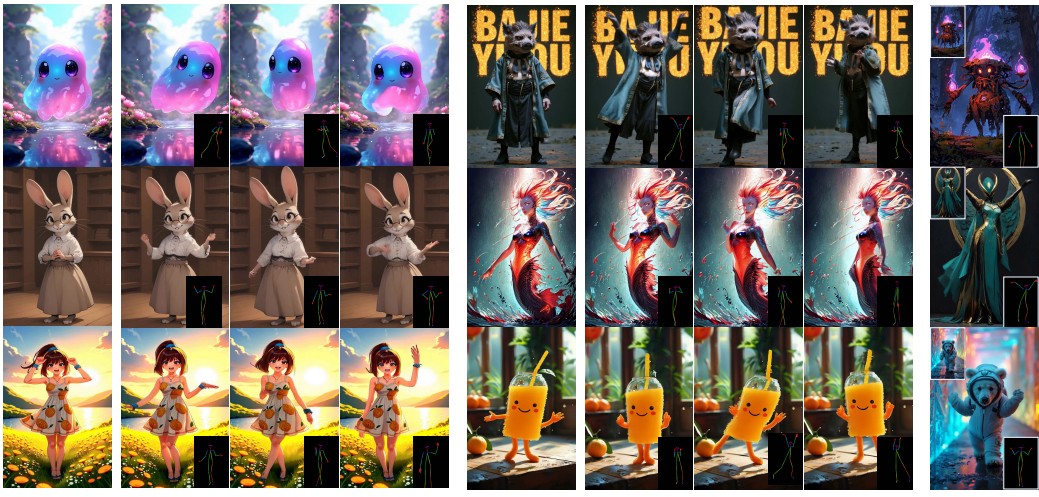

Figure 1: Animations produced by `Animate-X` which extends beyond human to anthropomorphic characters with various body structures, *e.g.*, without limbs, from games, animations, and posters.

## ABSTRACT

Character image animation, which generates high-quality videos from a reference image and target pose sequence, has seen significant progress in recent years. However, most existing methods only apply to human figures, which usually do not generalize well on anthropomorphic characters commonly used in industries like gaming and entertainment. Our in-depth analysis suggests to attribute this limitation to their insufficient modeling of motion, which is unable to comprehend the movement pattern of the driving video, thus imposing a pose sequence rigidly onto the target character. To this end, this paper proposes `Animate-X`, a universal animation framework based on LDM for various character types (collectively named `X`), including anthropomorphic characters. To enhance motion representation, we introduce the Pose Indicator, which captures comprehensive motion pattern from the driving video through both implicit and explicit manner. The former leverages CLIP visual features of a driving video to extract its gist of motion, like the overall movement pattern and temporal relations among motions, while the latter strengthens the generalization of LDM by simulating possible inputs in advance that may arise during inference. Moreover, we introduce a new Animated Anthropomorphic Benchmark ($A^2$`Bench`) to evaluate the performance of `Animate-X` on universal and widely applicable animation images. Extensive experiments demonstrate the superiority and effectiveness of `Animate-X` compared to state-of-the-art methods.

---

∗ Work done during internship at Ant Group.

† Project lead and corresponding author.

# 1 INTRODUCTION

Character image animation Yang et al. (2018); Zablotskaia et al. (2019b) is a compelling and challenging task that aims to generate lifelike, high-quality videos from a reference image and a target pose sequence. A modern image animation method shall ideally *balance* the identity preservation and motion consistency, which contributes to the promise of broad utilization Hu et al. (2023); Xu et al. (2023a); Chang et al. (2023a); Jiang et al. (2022). The phenomenal successes of GAN Goodfellow et al. (2014); Yu et al. (2023); Zhang et al. (2022b) and generative diffusion models Ho et al. (2022; 2020); Guo et al. (2023) have reshaped the performance of character animation generation. Nevertheless, most existing methods only apply to the human-specific character domain. In practice, the concept of *"character"* encompasses a much broader concept than human, including anthropomorphic figures in cartoons and games, collectively referred to as X, which are often more desirable in gaming, film, short videos, etc. The difficulty in extending current models to these domains can be attributed to two main factors: (1) the predominantly human-centered nature of available datasets, and (2) the limited generalization capabilities of current motion representations.

The limitations are clearly evidenced for non-human characters in Fig. 5. To replicate the given poses, the diffusion models trained on human dance video datasets tend to introduce unrelated human characteristics which may not make sense to reference figures, resulting in abnormal distortions. In other words, these models treat identity preservation and motion consistency as *conflicting* goals and struggle to balance them, while motion control often prevails. This issue is particularly pronounced for non-human anthropomorphic characters, whose body structures often differ from human anatomy—such as disproportionately large heads or the absence of arms, as shown in Fig. 1. The primary cause is that the motion representations extracted merely from pose conditions are hard to generalize to a broad range of common cartoon characters with unique physical characteristics, leading to their excessive sacrifices in identity preservation in favor of strict pose consistency, which is an unsensible trade-off between these *conflicting* goals.

To address this issue, the natural approach is to enhance the flexibility of motion representations without discarding current pose condition, which can prevent the model from making unsensible trade-offs between overly precise poses and low fidelity to reference images. To this end, we identify two key limitations of existing methods. **First**, the simple 2D pose skeletons, constructed by connecting sparse keypoints, lack of image-level details and therefore cannot capture the essence of the reference video, such as motion-induced deformations (e.g., body part overlap and occlusion) and overall motion patterns. **Second**, the self-driven reconstruction strategy aligns reference and pose skeletons by body shape, simplifying animation but ignoring shape differences during inference. These inspire us to design the new Pose Indicator from both implicit and explicit perspectives.

In this paper, we propose Animate-X for animating any character X. Sparked by generative diffusion models Rombach et al. (2022), we employ a 3D-UNet Blattmann et al. (2023) as the denoising network and provide it with motion feature and figure identity as condition. To fully capture the gist of motion from the driving video, we introduce the Pose Indicator, which consists of the Implicit Pose Indicator (IPI) and the Explicit Pose Indicator (EPI). Specifically, IPI extracts implicit motion-related features with the assistance of CLIP image feature, isolating essential motion patterns and relations that cannot be directly represented by the pose skeletons from the driving video. Meanwhile, EPI enhances the representation and understanding of the pose encoder by simulating real-world misalignments between the reference image and driven poses during training, strengthening the ability to generate explicit pose features. With the combined power of implicit and explicit features, Animate-X demonstrates strong character generalization and pose robustness, enabling general X character animation even though it is trained solely on human datasets. Moreover, we introduce a new **A**nimated **A**nthropomorphic **Bench**mark ($A^2$Bench), which includes 500 anthropomorphic characters along with corresponding dance videos, to evaluate the performance of Animate-X on other types of characters. Extensive experiments on both public human animation datasets and $A^2$Bench demonstrate that Animate-X outperforms state-of-the-art methods in preserving identity and maintaining motion consistency in animating X. Main contributions summarized as follows:

- We present Animate-X, which facilitates image-conditioned pose-guided video generation with high generalizability, particularly for attractive anthropomorphic characters. To the best of our knowledge, this is the first work to animate generic cartoon images without the need for strict pose alignment.

- The rethinking about the motion inspire us to propose Pose Indicator, which extracts motion representation suitable for anthropomorphic characters in both implicit and explicit manner, enhancing the robustness of `Animate-X`.

- Since the popular datasets only contain human video with limited character diversity, we present a new $A^2$`Bench`, specifically for evaluating performance on anthropomorphic characters. Extensive experiments demonstrate that our `Animate-X` outperforms the competing methods quantitatively and qualitatively on both $A^2$`Bench` and current human animation benchmark.

## 2 RELATED WORK

### 2.1 DIFFUSION MODELS FOR IMAGE/VIDEO GENERATION

In recent years, diffusion models Song et al. (2021); Ho et al. (2020) have demonstrated strong generative capabilities, pushing image generation technique towards a daily productivity tool Nichol et al. (2022); Ramesh et al. (2022); Mou et al. (2023); Huang et al. (2023); Zhang et al. (2023a); Liu et al. (2023). Pioneering works such as DALL-E 2 Ramesh et al. (2022) and Imagen Saharia et al. (2022) have showcased the extraordinary potential of diffusion models for high-quality image synthesis. Notable contributions, including Stable Diffusion Rombach et al. (2022), have well balanced scalability and efficiency, making diffusion-based image generation accessible and versatile across various applications. On the video generation front, diffusion models are making amazing progress Singer et al. (2023); Wang et al. (2023a; 2024c); Wu et al. (2023); Chai et al. (2023); Ceylan et al. (2023); Guo et al. (2023); Zhou et al. (2022); An et al. (2023); Xing et al. (2023); Qing et al. (2023); Yuan et al. (2023); Tan et al. (2024e); Gong et al. (2024); Wei et al. (2024a;b); Tan et al. (2024a); Shi et al. (2024). These methods joint spatio-temporal modeling to generate realistic motion dynamics and ensure temporal consistency, marking a substantial step forward in generative models for video content. In this work, we aim to tackle the character-centered image animation task, a dedicated of conditional video generation. Our approach enables the transformation of static images into dynamic animations by conditioning on desired motion. This innovation bridges the gap between image and video generation, highlights the versatility and adaptability of diffusion models in creating engaging visual narratives.

### 2.2 POSE-GUIDED CHARACTER MOTION TRANSFER

Character image animation aims to transfer motion from the source character to the target identity Zhang et al. (2024); Chang et al. (2023b), which has experienced an impressive journey to improve animation quality and versatility. Early works Li et al. (2019); Siarohin et al. (2019b; 2021b); Zhao & Zhang (2022b); Tan et al. (2024b); Wang et al. (2022); Tan et al. (2024d;c; 2023); Pan et al. (2024) predominantly utilize Generative Adversarial Networks (GANs) to generate animated human images. However, these GAN-based models are often confronted by the emergence of various artifacts in the generated outputs. With the advent of diffusion models, researchers Shen et al. (2024); Zhu et al. (2024) explored how to go beyond GANs. One effort is Disco Wang et al. (2023b), which leverages ControlNet Zhang et al. (2023b) to facilitate human dance generation, demonstrating the potential of diffusion models in generating dynamic human poses. Following this, MagicAnimate Xu et al. (2023b) and Animate Anyone Hu et al. (2023) introduce transformer-based temporal attention modules Vaswani (2017), enhancing the temporal consistency of animations and resulting in more smooth movement transitions. Sparked by the linear time efficiency of Mamba Gu & Dao (2023); Gu et al. (2021) conceptually merges the merits of parallelism and non-locality, Unianimate Wang et al. (2024b) resorts to it resorts to Mamba for efficient temporal modeling.

While these approaches have improved the realism of the animations, a notable limitation remains: most current methods require strict alignment between a reference image and driving video. This restricts their applicability in the scenarios where poses cannot be easily extracted, such as anthropomorphic characters, often resulting in bizarre and unsatisfactory outputs. In contrast, our approach adopts a robust and flexible motion representation to mitigate the dependence on pose alignment. This enables the generation of high-quality animations even in cases where previous methods struggle with non-alignable poses. In this manner, our method enhances the versatility and applicability of character image animation across a broad range of contexts (`X` character).

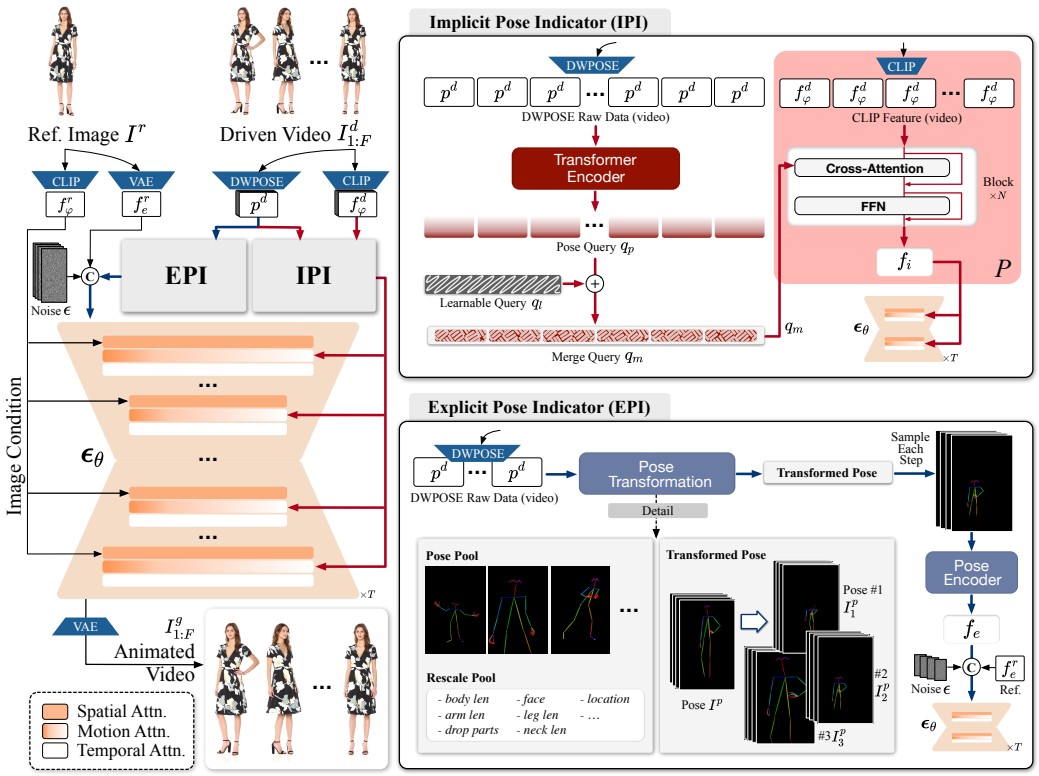

Figure 2: (a) The overview of our Animate-X. Given a reference image $I^r$, we first extract CLIP image feature $f_\varphi^r$ and latent feature $f_e^r$ via CLIP image encoder $\Phi$ and VAE encoder $\mathcal{E}$. The proposed Implicit Pose Indicator (**IPI**) and Explicit Pose Indicator (**EPI**) produce motion feature $f_i$ and pose feature $f_e$, respectively. $f_e$ is concatenated with the noised input $\epsilon$ along the channel dimension, then further concatenated with $f_e^r$ along the temporal dimension. This serves as the input to the diffusion model $\epsilon_\theta$ for progressive denoising. During the denoising process, $f_\varphi^r$ and $f_i$ provide appearance condition from $I^r$ and motion condition from $I_{1:F}^d$. At last, a VAE decoder $\mathcal{D}$ is adopted to map the generated latent representation $z_0$ to the animation video. (b) The detailed structure of Implicit Pose Indicator. (c) The pipeline of pose transformation by Explicit Pose Indicator.

## 3 METHOD

In this work, we aim to generate an animated video that maintains consistency in identity with a reference image $I^r$ and body movement with a driving video $I_{1:F}^d$. Different from previous works, our primary objective is to animate a general characters beyond human, particularly like anthropomorphic ones, which has broader applications in entertainment industry.

### 3.1 PRELIMINARIES OF LATENT DIFFUSION MODEL

A diffusion model (DM) operates by learning a probabilistic process that models data generation through noise. To mitigate the heavy computational load of traditional pixel-based diffusion models in high-dimensional RGB spaces, latent diffusion models (LDMs) Rombach et al. (2022) propose to shift the process into a lower-dimensional latent space using a pre-trained variational autoencoder (VAE) Kingma (2013). It encodes the input data into a compressed latent representation $z_0$. Gaussian noise is then incrementally added to this latent representation over several steps, reducing computational requirements while maintaining the generative capabilities of the model. The process can be formalized as:

$$q(\mathbf{z}_t|\mathbf{z}_{t-1}) = \mathcal{N}(\mathbf{z}_t; \sqrt{1-\beta_t}\mathbf{z}_{t-1}, \beta_t\mathbf{I}), \tag{1}$$

where $\beta_t \in (0,1)$ represents the noise schedule. As $t \in 1, 2, ..., \mathcal{T}$ increases, the cumulative noise applied to the original $\mathbf{z}_0$ intensifies, causing $\mathbf{z}_t$ to progressively resemble random Gaussian noise.

Compared to the forward diffusion process, the reverse denoising process $p_\theta$ aims to reconstruct the clean sample $\mathbf{z}_0$ from the noisy input $\mathbf{z}_t$. We represent the denoising step $p(\mathbf{z}t - 1|\mathbf{z}t)$ as follows:

$$p_\theta(\mathbf{z}_{t-1}|\mathbf{z}_t) = \mathcal{N}(\mathbf{z}_{t-1}; \boldsymbol{\mu}_\theta(\mathbf{z}_t, t), \boldsymbol{\Sigma}_\theta(\mathbf{z}_t, t)), \tag{2}$$

in which $\boldsymbol{\mu}_\theta(\mathbf{z}_t, t)$ refers to the estimated target of the reverse diffusion process and the process typically is achieved by a diffusion model $\boldsymbol{\epsilon}_\theta$ with the parameters $\theta$. To model the temporal dimension, the denoising model $\boldsymbol{\epsilon}_\theta$ is commonly built on a 3D-UNet architecture Blattmann et al. (2023) in video generation methods Hu et al. (2023); Wang et al. (2023c). Given the input conditional guidance $c$, they usually use an L2 loss to reduce the difference between the predicted noise and the ground-truth noise during the optimization process:

$$\mathcal{L} = \mathbb{E}_\theta \left[ \|\boldsymbol{\epsilon} - \boldsymbol{\epsilon}_\theta(\mathbf{z}_t, t, c)\|^2 \right] \tag{3}$$

once the reversed denoising stage is complete, the predicted clean latent is passed through the VAE decoder to reconstruct the predicted video in pixel space.

### 3.2 POSE INDICATOR

To extract motion representations, previous works typically detect the pose keypoints via DW-Pose Yang et al. (2023) from the driven video $I_{1:F}^d$ and further visualize them as pose image $I^p$, which are trained using self-driven reconstruction strategy. However, it brings several limitations as mentioned in Sec. 1: (1) The sole pose skeletons lack image-level details and are therefore unable to capture the essence of the reference video, such as motion-induced deformations and overall motion patterns. (2) The self-driven reconstruction training strategy naturally aligns the reference and pose images in terms of body shape, which simplifies the animation task by overlooking likely body shape differences between the reference image and the pose image during inference. Both limitations weaken the model to develop a deep, holistic motion understanding, leading to **inadequate** motion representation. To address these issues, we propose Pose Indicator, which consists of Implicit Pose Indicator (IPI) and Explicit Pose Indicator (EPI).

**Implicit Pose Indicator (IPI).** To extract unified motion representations from the driving video in the first limitation, we resort to the CLIP image feature $f_\varphi^d = \Phi(I_{1:F}^d)$ extracted by a CLIP Image Encoder. CLIP utilizes contrastive learning to align the embeddings of related images and texts, which may include descriptions of appearance, movement, spatial relationships and etc. Therefore, the CLIP image feature is actually a highly entangled representation, containing motion patterns and relations helpful to animation generation. As presented in Fig. 2 (a), we introduce a lightweight extractor $P$ which is composed of $N$ stacked layers of cross-attention and feed-forward networks (FFN). In cross attention layer, we employ $f_\varphi^d$ as the keys ($K$) and values ($V$). Consequently, the challenge becomes designing an appropriate query ($Q$), which should act as a guidance for motion extraction. Considering that the keypoints $p^d$ extracted by DWPose provide a direct description of the motion, we design a transformer-based encoder to obtain the embedding $q_p$, which is regarded as an ideal candidate for $Q$. Nevertheless, motion modeling using sole sparse keypoints is overly simplistic, resulting in the loss of underlying motion patterns. To this end, we draw inspiration from query transformer architecture Awadalla et al. (2023); Jaegle et al. (2021) and initialize a learnable query vector $q_l$ to complement sparse keypoints. Subsequently, we feed the merged query $q_m = q_p + q_l$ and $f_\varphi^d$ into $P$ and get the implicit pose indicator $f_i$, which contains the essential representation of motion that cannot be represented by the simple 2D pose skeletons.

**Explicit Pose Indicator (EPI).** To deal with the second limitation in the training strategy, we propose EPI, designed to train the model to handle misaligned input pairs during inference. The *key insight* lies in simulating misalignments between reference image and pose images during training while ensuring the motion remains consistent with the given driving video $I_{1:F}^d$. Therefore, we explore two pose transformation schemes: Pose Realignment and Pose Rescale. As shown in Fig. 2 (b), in the pose realignment scheme, we first establish a pose pool containing pose images from the training set. In each training step, we first sample the reference image $I^r$ and the driving pose $I^p$ following previous works. Additionally, we randomly select an align anchor pose $I_{anchor}^p$ from the pose pool. This anchor serves as a reference for aligning the driving pose, producing the aligned pose $I_{realign}^p$. However, since the characters we aim to animate are often anthropomorphic characters, whose shapes can significantly differ from human, such as varying head-to-shoulder ratios, extremely short legs, or even the absence of arms (as shown in Fig. 1 and Fig. 5), relying solely

on pose realignment is insufficient to capture these variations for simulation. Therefore, we further introduce Pose Rescale. Specifically, we define a set of keypoint rescaling operations, including modifying the length of the body, legs, arms, neck, and shoulders, altering face size, even adding or removing specific body parts and etc. These transformations are stored in a rescale pool. After obtaining the realigned poses $I_{realign}^p$, we apply a random selection of transformations from this pool with a certain probability on them, generating the final transformed poses $I_n^p$ (additional examples of transformations are provided in the Appendix A). Note that we set the probability of $\lambda \in [0, 1]$ to apply the pose transformation, and with a probability of $1 - \lambda$, the pose image remains unchanged. Subsequently, $I_n^p$ is encoded to the explicit feature $f_e$ via a Pose Encoder.

## 3.3 FRAMEWORK AND IMPLEMENT DETAILS

In light of the success of previous works Hu et al. (2023); Zhang et al. (2024), Animate-X follows the main framework, which consists of several encoders for feature extraction and a 3D-UNet Wang et al. (2023a;c); Blattmann et al. (2023) for video generation. As shown in Fig. 2, given a reference image $I^r$, we employ the pretrained CLIP Image Encoder $\Phi$ Radford et al. (2021) to extract appearance feature $f_\varphi^r$ from $I^r$. To reduce the parameters of the framework and facilitate appearance alignment, we exclude the Reference Net presented in most of the previous works Hu et al. (2023); Zhang et al. (2024); Zhu et al. (2024). Instead, a VAE encoder $\mathcal{E}$ is utilized to extract the latent representation $f_e^r$ from $I^r$, which is then directly used as part of the input for the denoising network $\epsilon_\theta$ following Wang et al. (2024b). For the driven video $I_{1:F}^d$, we detect the pose keypoints $p^d$ and CLIP feature $I^d$ via a DWPose Yang et al. (2023) and CLIP Image Encoder $\Phi$. Subsequently, IPI and EPI introduced in Sec. 3.2 extract the implicit latent $f_i$ and explicit latent $f_e$, respectively. The explicit $f_e$ is first concatenated with the noised latent $\epsilon$ to obtain the fused features along the channel dimension, which is further stacked with $f_e^r$ along the temporal dimension, resulting in combined features $f_{merge}$. Then, the combined features are fed into the video diffusion model $\epsilon_\theta$ for jointly appearance alignment and motion modeling. The diffusion model $\epsilon_\theta$ comprises multiple stacked layers of Spatial Attention, Motion Attention and Temporal Attention. The Spatial Attention receives inputs from $f_{merge}$ and $f_i^r$ and fuses the identity condition from $I^r$ with the motion condition from $I^d$ through cross-attention (CA), producing an intermediate representation $x$. To further enhance motion consistency, the implicit representation $f_i$ is fed into the Motion Attention module, along with $x$ in the form of a residual connection, resulting in the representation $x' = x + \mathrm{CA}(x, f_i)$. Inpsired by the linear time efficiency of Mamba Gu & Dao (2023) in long sequence processing, we employ it as Temporal Attention module to maintain the temporal consistency.

**Training and Inference.** To improve the model's robustness against pose and reference image misalignments, we adopt two key training schemes. First, we set a high transformation probability $\lambda$ (over 98%) in the EPI, enabling the model to handle a wide range of misalignment scenarios. Second, we apply random dropout to the input conditions at a predefined rate Wang et al. (2024b). After that, while the reference image and driven video are from the same human dancing video during training, in the inference phase (Fig. 9 (b)), Animate-X can handle an arbitrary reference image and driven video, which may differ in appearance.

## 3.4 $A^2$BENCH

The main task of our Animate-X is to animate an anthropomorphic character with vivid and smooth motions. However, current publicly available datasets Jafarian & Park (2021); Zablotskaia et al. (2019a) primarily focus on human animation and fall short in capturing a broad range of anthropomorphic characters and corresponding dancing videos. This gap makes these datasets and benchmarks unsuitable for quantitatively evaluating different methods in anthropomorphic character animation.

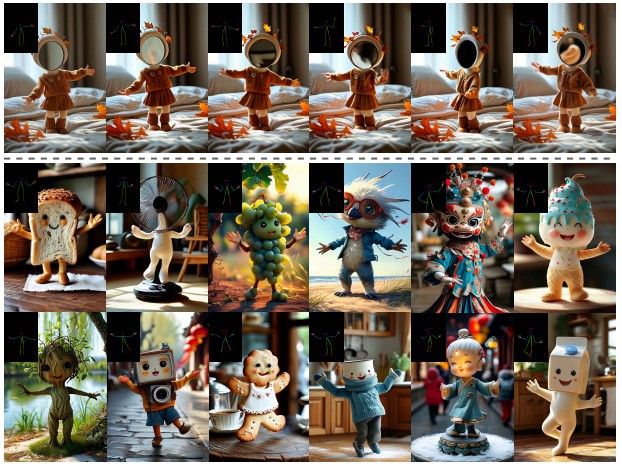

Figure 3: Examples from our $A^2$Bench.

| Method | PSNR* ↑ | SSIM ↑ | L1 ↑ | LPIPS ↓ | FID ↓ | FID-VID ↓ | FVD ↓ |
|---|---|---|---|---|---|---|---|
| Moore-AnimateAnyone Corporation (2024) | 9.86 | 0.299 | 1.58E-04 | 0.626 | 50.97 | 75.11 | 1367.84 |
| MimicMotion Zhang et al. (2024) (ArXiv24) | 10.18 | 0.318 | 1.51E-04 | 0.622 | 122.92 | 129.40 | 2250.13 |
| ControlNeXt Peng et al. (2024) (ArXiv24) | 10.88 | 0.379 | 1.38E-04 | 0.572 | 68.15 | 81.05 | 1652.09 |
| MusePose Tong et al. (2024) (ArXiv24) | 11.05 | 0.397 | 1.27E-04 | 0.549 | 100.91 | 114.15 | 1760.46 |
| Unianimate Wang et al. (2024b) (ArXiv24) | 11.82 | 0.398 | 1.24E-04 | 0.532 | 48.47 | 61.03 | 1156.36 |
| **Animate-X** | **13.60** | **0.452** | **1.02E-04** | **0.430** | **26.11** | **32.23** | **703.87** |

Table 1: Quantitative comparisons with SOTAs on $A^2$Bench with the rescaled pose setting. "PSNR*" means using the modified metric Wang et al. (2024a) to avoid numerical overflow.

| Method | PSNR* ↑ | SSIM ↑ | L1 ↑ | LPIPS ↓ | FID ↓ | FID-VID ↓ | FVD ↓ |
|---|---|---|---|---|---|---|---|
| FOMM Siarohin et al. (2019a) (NeurIPS19) | 10.49 | 0.363 | 1.47E-04 | 0.613 | 183.18 | 147.82 | 2535.12 |
| MRAA Siarohin et al. (2021a) (CVPR21) | 12.62 | 0.420 | 1.09E-04 | 0.556 | 161.57 | 196.87 | 3094.68 |
| LIA Wang et al. (2022) (ICLR22) | 13.78 | 0.445 | 9.70E-05 | 0.497 | 105.13 | 78.51 | 1813.28 |
| DreamPose Karras et al. (2023) (ICCV23) | 7.76 | 0.305 | 2.28E-04 | 0.534 | 277.64 | 315.58 | 4324.42 |
| MagicAnimate Xu et al. (2023a) (CVPR24) | 11.90 | 0.396 | 1.17E-04 | 0.523 | 117.09 | 117.54 | 2021.93 |
| Moore-AnimateAnyone Corporation (2024) (CVPR24) | 11.56 | 0.360 | 1.27E-04 | 0.532 | 37.82 | 59.80 | 1117.29 |
| MimicMotion Zhang et al. (2024) (ArXiv24) | 12.66 | 0.407 | 1.07E-04 | 0.497 | 96.46 | 61.77 | 1368.83 |
| ControlNeXt Peng et al. (2024) (ArXiv24) | 12.82 | 0.421 | 1.02E-04 | 0.472 | 46.66 | 59.41 | 1152.96 |
| MusePose Tong et al. (2024) (ArXiv24) | 12.92 | 0.438 | 9.90E-05 | 0.470 | 80.22 | 87.97 | 1401.96 |
| **Animate-X** | **14.10** | **0.463** | **8.92E-05** | **0.425** | **31.58** | **33.15** | **849.19** |

Table 2: Quantitative comparisons with existing methods on $A^2$Bench in the self-driven setting. Underline means the second best result.

To bridge this gap, we propose the **A**nimated **A**nthropomorphic character **Bench**mark ($A^2$Bench) to comprehensively evaluate the performance of different methods. Specifically, we first provide a prompt template to GPT-4 OpenAI (2024) and leverage it to generate 500 prompts, each of which contains a textual description of an anthropomorphic character. Please refer to Appendix B.2 for details. Inspired by the powerful image generation capability of KLing AI Technology (2024), we feed the produced prompts into its Text-To-Image module, which synthesizes the corresponding anthropomorphic character images according to the given text prompts. Subsequently, the Image-To-Video module is employed to further make the characters in the images dance vividly. For each prompt, we repeat the process for 4 times and filter the most satisfactory image-video pairs as the output corresponding to this prompt. In this manner, we collect 500 anthropomorphic characters and the corresponding dance videos, as shown in Fig. 3. Please refer to Appendix B for details.

## 4 EXPERIMENTS

### 4.1 EXPERIMENTAL SETTINGS

**Dataset.** We collect approximately 9,000 human videos from the internet and supplement this with TikTok dataset Jafarian & Park (2021) and Fashion dataset Zablotskaia et al. (2019a)

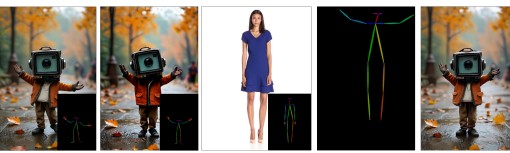

Ref Image $I^a$    Driven pose $P^a$    Align target $P^b$    Aligned pose $P_b^a$    Our result

Figure 4: The illustration of comparison settings.

for training. Following previous works Hu et al. (2023); Zablotskaia et al. (2019a); Jafarian & Park (2021), we use 10 and 100 videos for both qualitative and quantitative comparisons from TikTok and Fashion dataset, respectively. We additionally experimented on 100 image-video pairs selected from the newly proposed $A^2$Bench introduced in Sec 3.4. Please note that, to ensure a fair comparison, the data in the $A^2$Bench are **not** included in the training set to train our model. The data are only used to evaluate the quantitative results and provide interesting reference image cases.

**Evaluation Metrics.** We assess the results using evaluation metrics in Appendix B.1, including PSNR Hore & Ziou (2010), SSIM Wang et al. (2004), L1, LPIPS Zhang et al. (2018), which are widely-used image metrics for measuring the visual quality of the generated results. In addition, we introduce FID Heusel et al. (2017), FID-VID Balaji et al. (2019) and FVD Unterthiner et al. (2018) to quantify the discrepancy between the generated video distribution and the real video distribution.

### 4.2 EXPERIMENTAL RESULTS

**Quantitative Results.** Since our Animate-X primarily focuses on animating the anthropomorphic characters, very few of which, if not none, can be extracted the pose skeleton accurately by

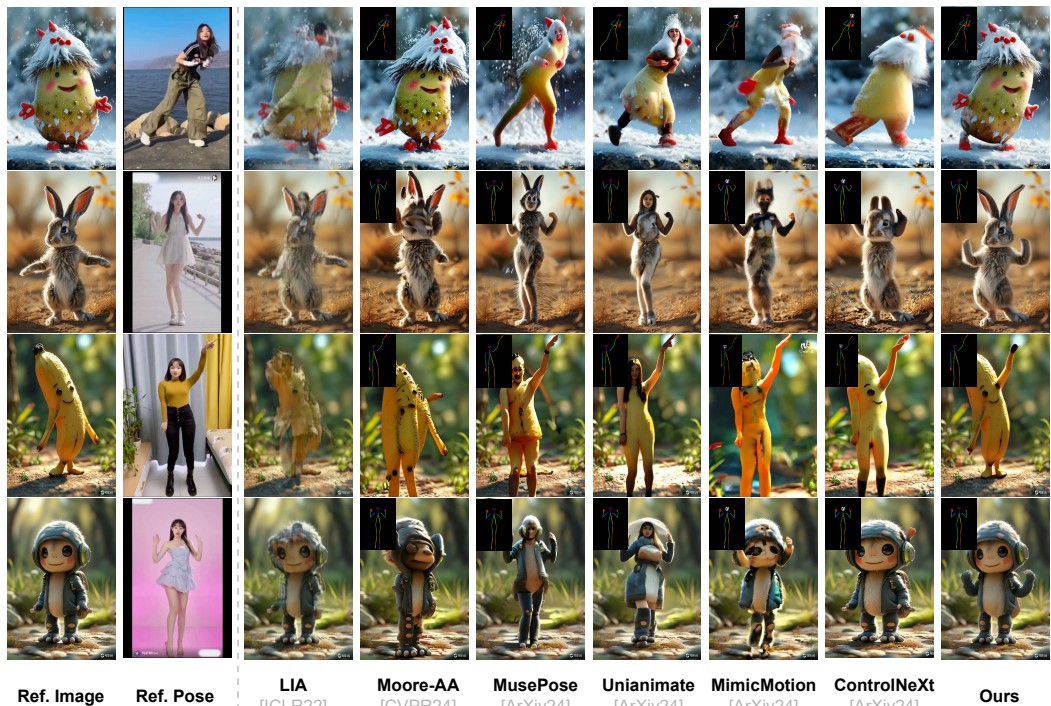

Figure 5: Qualitative comparisons with state-of-the-art methods.

DWPose Yang et al. (2023). It naturally leads to a misalignment of the input reference image with the driving pose images. To compute quantitative results in this case, we set up a new comparison setting. For each case in $A^2$Bench (*i.e.*, a reference image $I^a$ and a pose $P^a$, as shown in Fig. 4), we randomly select one human's pose image $P^b$ and align the anthropomorphic character's pose $P^a$ to it, such that the aligned pose $p_b^a$ retains the movements of $P^a$ but has the same body shape (fat/thin, tall/short, *etc.*) as $p^b$. Ultimately, we take the anthropomorphic character $I_a$ and the aligned driving pose image $p_b^a$ as inputs to the model, generating results that allow it to calculate quantitative metrics with the original anthropomorphic character dancing video in $A^2$Bench. In this setting, we compare our method with Animate Anyone Hu et al. (2023), Unianimate Wang et al. (2024b), MimicMotion Zhang et al. (2024), ControlNeXt Peng et al. (2024) and MusePose Tong et al. (2024), which also use pose images (*e.g.*, $P^b$ in Fig. 4) as input. The results of Animate Anyone Hu et al. (2023) are obtained by leveraging the publicly available reproduced code Corporation (2024). Tab. 1 presents the quantitative results, where Animate-X markedly surpasses all comparative methods in terms of all metrics. It is worth noting that, we do not use $A^2$Bench as training data to avoid overfitting and ensure fair comparisons, in line with other comparative methods.

Following previous works which evaluate quantitative results in self-driven and reconstruction manner, we additionally compare our method with (a) GAN-based image animate works: FOMM Siarohin et al. (2019a), MRAA Siarohin et al. (2021a), LIA Wang et al. (2022). (b) Diffusion model-based image animate works: DreamPose Karras et al. (2023), MagicAnimate Xu et al. (2023a) and present the results in Tab. 2, which indicates that our method achieves the best performance across all the metrics. Moreover, we provide the quantitative results on the human dataset (TikTok and Fashion) in Tab. 10 and Tab. 11, respectively. Please refer to Appendix D.2 for details. Animate-X reaches the comparable score to Unianimate and exceeds other SOTA methods, which demonstrates the superiority of Animate-X on **both** anthropomorphic and human benchmarks.

**Qualitative Results.** Qualitative comparisons of anthropomorphic animation are shown in Fig. 5. We observe that GAN-based LIA Wang et al. (2022) does not generalize well, which can only work on a specific dataset like Siarohin et al. (2019b). Benefiting from the powerful generative capabilities of the diffusion model, Animate Anyone Hu et al. (2023) renders a higher resolution image, but the identity of the image changes and do not generate an accurate reference pose motion. Although MusePose Tong et al. (2024), Unianimate Wang et al. (2024b) and MimicMotion Zhang et al. (2024) improve the accuracy of the motion transfer, these methods generate a unseen person, which is not

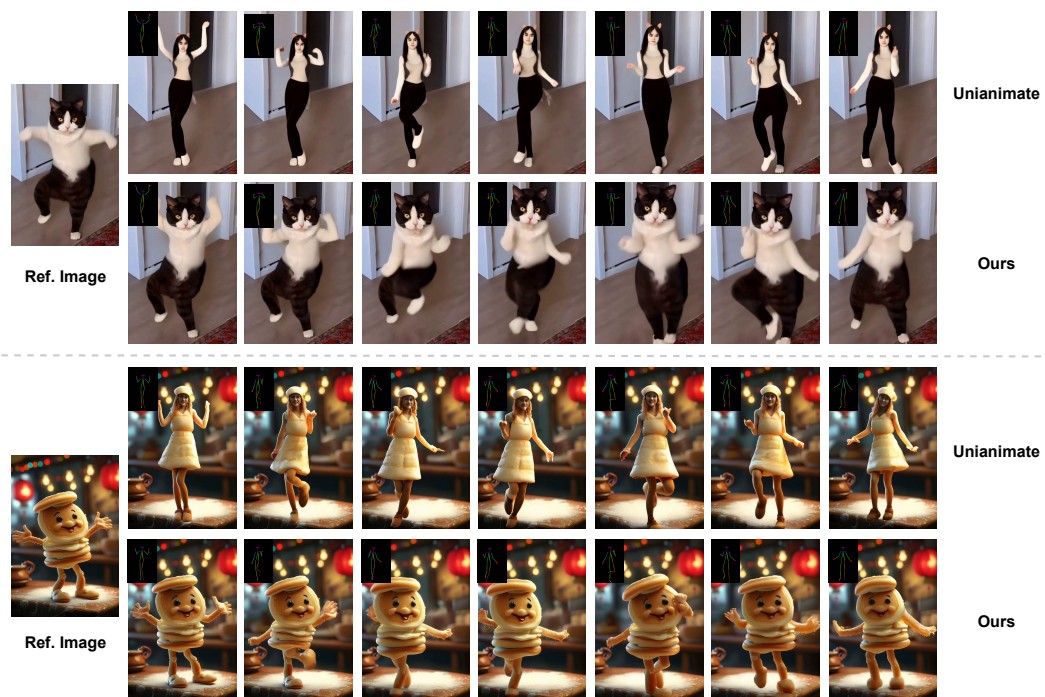

Figure 6: Qualitative comparisons with Unianimate in terms of long video generation.

| Method | Moore-AA | MimicMotion | ControlNeXt | MusePose | Unianimate | **Animate-X** |
|---|---|---|---|---|---|---|
| Identity preservation ↑ | 60.4% | 14.8% | 52.0% | 31.3% | 43.0% | **98.5%** |
| Temporal consistency ↑ | 19.8% | 24.9% | 36.9% | 43.9% | 81.1% | **93.4%** |
| Visual quality ↑ | 27.0% | 17.2% | 40.4% | 40.3% | 79.3% | **95.8%** |

Table 3: User study results.

the desired result. ControlNeXt combines the advantages of the above two types of methods, so maintains the consistency of identity and motion transfer to some extent, yet the results are somewhat unnatural and unsatisfactory, *e.g.*, the ears of the rabbit and the legs of the banana in Fig. 5. In contrast, Animate-X ensures both identity and consistency with the reference image while generating expressive and exaggerated figure motion, rather than simply adopting quasi-static motion of the target character. Further, we present some long video comparisons in Fig. 6. Unianimate generates a woman out of thin air who dances according to the given pose images. Animate-X animates the reference image in a cute way while preserving appearance and temporal continuity, and it does not generate parts that do not originally exist. In summary, Animate-X excels in maintaining appearance and producing precise, vivid animations with a high temporal consistency. Please refer to Appendix D.1 for details.

**User Study.** To estimate the quality of our method and SOTAs from human perspectives, we conduct a blind user study with 10 participants. Specifically, we randomly select 10 characters from $A^2$Bench and collect 10 driving video from the website. For each of 6 methods tested, 10 animation clips are generated, resulting in a total of 60 clips. Each participant is presented two results generated by different methods for the same set of inputs and asked to choose which one is better in terms of *visual quality*, *identity preservation*, and *temporal consistency*. This process is repeated $C_2^6$ times. The results are summarized in Tab. 3, where our method noticeably outperforms other methods in all aspects, demonstrating its superiority and effectiveness. Details in Appendix C.

## 4.3 ABLATION STUDY

**Ablation on Implicit Pose Indicator.** To analyze the contributions of Implicit Pose Indicator, we remove it from Animate-X as w/o IPI and compare it with Baseline and Animate-X. From the first row of Fig. 7, we observe that Baseline generates a person whose appearance is appreciably distinct from the reference image. With the help of EPI, this problem is mildly mitigated. However, due to the absence of IPI, compared to Ours, there are still strange things and human-like hands

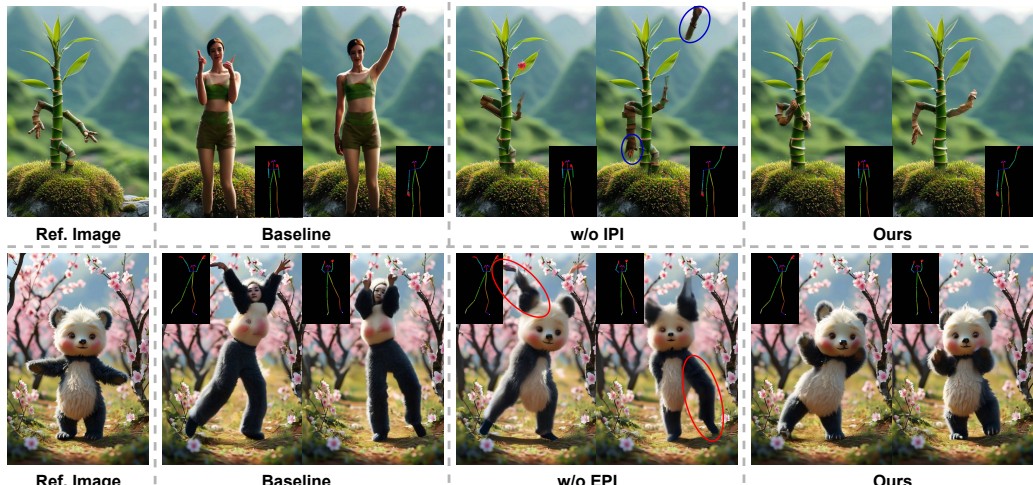

Figure 7: Visualization of ablation study on IPI and EPI.

appearing, as indicated by the blue circle. For more detailed analysis about the structure of IPI, we set up several variants: (1) remove IPI: w/o IPI. (2) remove learnable query: w/o LQ. (3) remove DWPose query: w/o DQ. The quantitative results are shown in Tab. 4. It can be seen that removing the entire IPI presents the worst performance. By modifying the IPI module, although it improves on the w/o IPI, it still falls short of the final result of `Animate-X`, which suggests that our current IPI structure is the most reasonable and achieves the best performance.

**Ablation on Explicit Pose Indicator.** We demonstrate the visual results of ablating EPI setting in the second row of Fig. 7 by removing EPI. Without EPI, although the appearance of the panda is preserved thanks to IPI, the model incorrectly treats the panda's ears as arms and forcibly stretches the legs to match

| Method | PSNR* ↑ | SSIM ↑ | L1 ↑ | LPIPS ↓ | FID ↓ | FID-VID ↓ | FVD ↓ |
|---|---|---|---|---|---|---|---|
| w/o IPI | 13.30 | 0.433 | 1.35E-04 | 0.454 | 32.56 | 64.31 | 893.31 |
| w/o LQ | 13.48 | 0.445 | 1.76E-04 | 0.454 | 28.24 | 42.74 | 754.37 |
| w/o DQ | 13.39 | 0.445 | **1.01E-04** | 0.456 | 30.33 | 62.34 | 913.33 |
| w/o EPI | 12.63 | 0.403 | 1.80E-04 | 0.509 | 42.17 | 58.17 | 948.25 |
| w/o Realign | 12.27 | 0.433 | 1.17E-04 | 0.434 | 34.60 | 49.33 | 860.25 |
| w/o Rescale | 13.23 | 0.438 | 1.21E-04 | 0.464 | 27.64 | 35.95 | 721.11 |
| **Animate-X** | **13.60** | **0.452** | 1.02E-04 | **0.430** | **26.11** | **32.23** | **703.87** |

Table 4: Quantitative results of ablation study.

the length of the legs in the pose image indicated by red circles. In contrast, these issues are completely resolved by the assistance of EPI. We further conduct more detailed ablation experiments for different pairs of pose transformations by (1) removing the entire EPI: w/o EPI. (3) remove Pose Realignment: w/o Realignment. (2) removing Pose Rescale: w/o Rescale; From the results displayed in Tab. 4, we found that Pose Realignment contributes the most. It suggests that simulating misalignment case in inference is the the key factor.

In summary, we can draw conclusions: (1) IPI facilitates the preservation of appearance and prevents the generation of content that does not exist in the reference image like human arms. (2) EPI prevents the forced alignment of a pose image that is not naturally aligned with the reference image during animation, thus avoiding the unintended animation of parts that should remain static like the panda's ears shown in Fig. 7. Please refer to Appendix D.5 for details.

## 5 CONCLUSIONS

In this study, we present `Animate-X`, a novel approach to character animation capable of generalizing across different types of characters named `X`. To address the imbalance between identity preservation and movement consistency caused by the insufficient motion representation, we introduce the Pose Indicator, which leverages both implicit and explicit features to enhance the motion understanding of the model. In this way, `Animate-X` demonstrates strong generalization and robustness, achieving general X character animation. The proposed framework showcases significant improvements over state-of-the-art methods in terms of identity preservation and motion consistency, as evidenced by experiments on both public datasets and the newly introduced $A^2$`Bench`, which features anthropomorphic characters. Limitation and ethical considerations see Appendix E.

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

# APPENDICES

## A    NETWORK DETAILS

Due to space constraints in the main paper, we only present a brief overview of the EPI process. Here, in Fig. 8, we provide a more detailed explanation of the pose transformation in EPI, along with additional case examples. First, we sample a driving pose $I^p$ and then randomly select an anchor pose $I^p_{anchor}$ from the pose pool (two examples are shown in Fig. 8). The driving pose $I^p$ is aligned to the anchor pose $I^p_{anchor}$, resulting in the aligned pose $I^p_{realign}$. Next, we apply several rescaling operations randomly chosen from the rescale pool to further modify the aligned pose $I^p_{realign}$. By combining different rescaling options, we can obtain multiple transformed poses $I^p_n$. However, it is important to note that in each training step, only one anchor pose $I^p_{anchor}$ and one rescaling combination are selected, so only one transformed pose $I^p_n$ is used for training. As shown in the Fig. 8, the transformed pose $I^p_n$ retains the same motion as the sampled pose $I^p$ but has a body shape similar to the anchor pose $I^p_{anchor}$. This simulates scenarios during inference where there are body shape differences between the reference image and the driving pose, enabling the model to generalize to such cases.

In the experiments, we use the visual encoder of the multi-modal CLIP-Huge model Radford et al. (2021) in Stable Diffusion v2.1 Rombach et al. (2022) to encode the CLIP embedding of the reference image and driving videos. The pose encoder, composed of several convolutional layers, follows a similar structure to the STC-encoder in VideoComposer Wang et al. (2023c). For model initialization, we employ a pre-trained video generation model Wang et al. (2024c), as done in previous approaches Xu et al. (2023a); Hu et al. (2023); Zhu et al. (2024); Wang et al. (2024b). The experiments are carried out using 8 NVIDIA A100 GPUs. During training, videos are resized to a spatial resolution of 768×512 pixels, and we feed the model with uniformly sampled video segments of 32 frames to ensure temporal consistency. We use the AdamW optimizer Loshchilov & Hutter (2017) with learning rates of 5e-7 for the implicit pose indicator and 5e-5 for other modules. For noise sampling, DDPM Ho et al. (2020) with 1000 steps is applied during training. In the inference phase, we adjust the length of the driving pose to align roughly with the reference pose and used the DDIM sampler Song et al. (2021) with 50 steps for faster sampling.

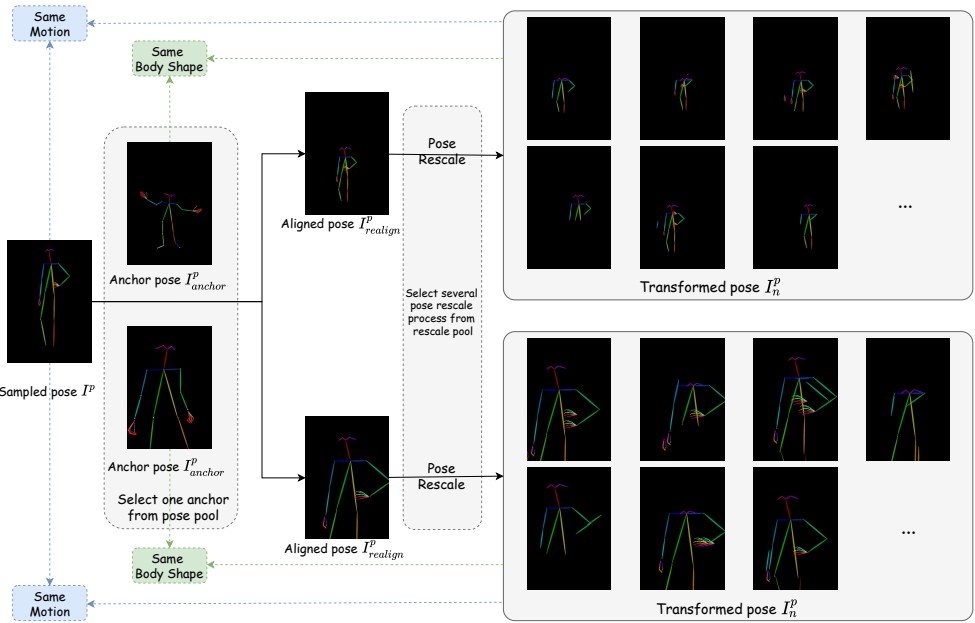

Figure 8: More example for EPI.

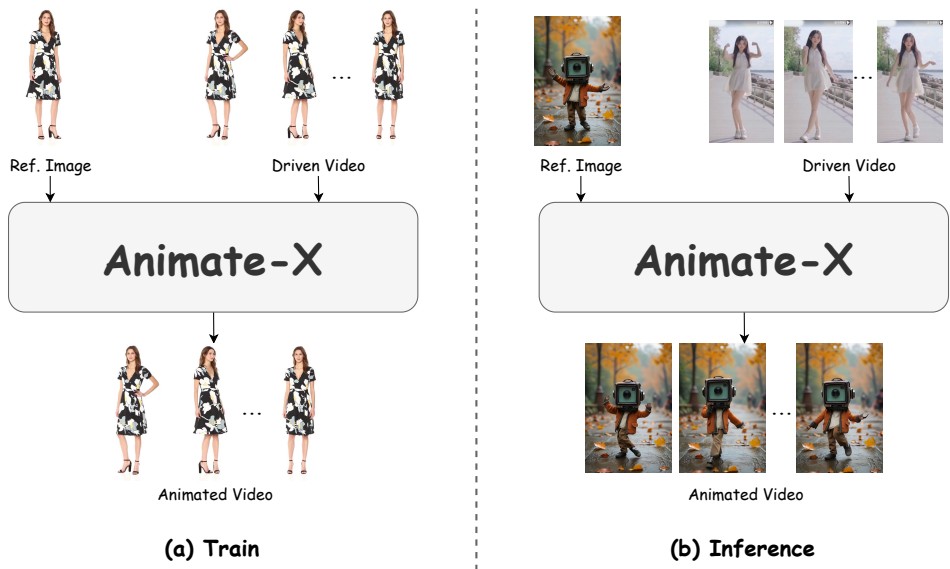

Figure 9: The difference of training and inference pipeline. During training, the reference image and the driven video come from the same video, while in the inference pipeline, the reference image and the driven video can be from any sources and appreciably different.

## B    BENCHMARK DETAILS

### B.1    EVALUATION METRIC

We employ several evaluation metrics to quantitatively assess our results, including PSNR, SSIM, L1, LPIPS, FID, FID-VID and FVD. The detailed metrics are introduced as follows:

- PSNR is a measure used to evaluate the quality of reconstructed images compared to the original ones. It is expressed in decibels (dB) and higher values indicate better quality. PSNR is commonly used in image compression and restoration fields.

- SSIM assesses the similarity between two images based on their luminance, contrast, and structural information. It considers perceptual phenomena affecting human vision and thus provides a better correlation with perceived image quality than PSNR.

- The L1 metric refers to the mean absolute difference between the corresponding pixel values of two images. It quantifies the average magnitude of errors in predictions without considering their direction, making it useful for measuring the extent of differences.

- LPIPS is a perceptual distance metric based on deep learning. It evaluates the similarity between images by analyzing the feature representations of image patches and tends to align well with human visual perception, making it suitable for tasks like image generation.

- FID is used to assess the quality of images generated by generative models (like GANs) by comparing the distribution of generated images to that of real images in feature space (extracted by a pretrained CNN). Lower FID values suggest that the generated images are more similar to real images.

- FID-VID extends the FID metric to video data. It measures the quality of generated videos by comparing the distribution of generated video features to real video features, providing insights into the temporal aspects of video generation.

- FVD is another metric for evaluating video generation, similar to FID. It measures the distance between the feature distributions of real and generated videos, taking both spatial and temporal dimensions into account. Lower FVD indicates that generated videos are closer to real ones regarding visual quality and dynamics.

Figure 10: Detailed pipeline for building $A^2$Bench based on large-scale pretrained models, including Open-ChatGPT 4o and KLing AI.

## B.2 Data Details

The detailed process for constructing $A^2$Bench is outlined in Fig. 10. We initially provide GPT-4o with a template that clearly specifies the demand to generate 'anthropomorphized' images. The images were required to be cute, with arms and legs, standing, dancing, and of high quality. To allow for a variety of image outputs, we left the fields for 'object', 'season', 'province', and 'specific

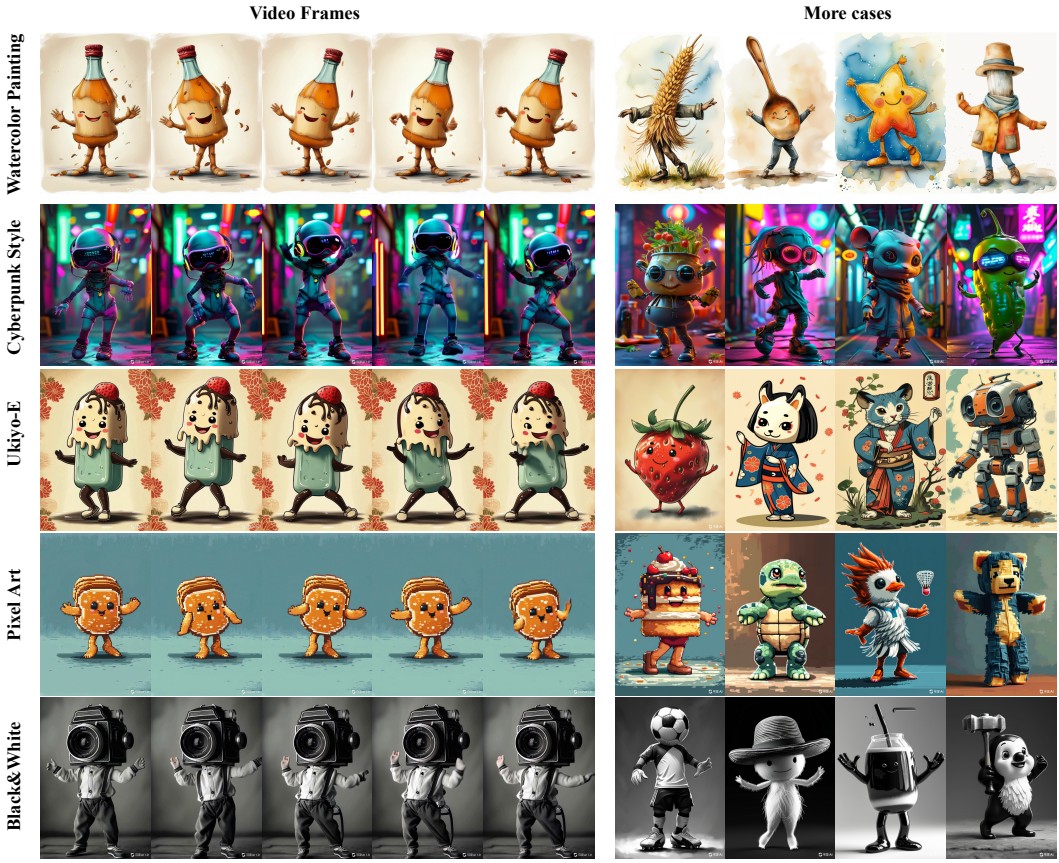

Figure 11: More styles in $A^2Bench$.

location' empty. For the key factor influencing diversity and relevance, i.e., 'object', we provide a selectable range, such as everyday items, furniture, fruits, and natural creatures. To help GPT-4o better understand our intent, we additionally provide two examples, where the prompts had already been proven to generate satisfactory images by text-to-image module of KLing AI. Thanks to the text understanding and generation capabilities of GPT-4o, we collect 500 prompts for image generation. We then fed these 500 prompts into the text-to-image module of Keling AI, obtaining corresponding anthropomorphic characters images. Based on these images, we further generate videos of them dancing using the image-to-video module of Keling AI. In this way, we collect 500 pairs of images and videos of anthropomorphic characters, forming our $A^2$Bench.

Moreover, we add style trigger words such as "*Watercolor Painting*", "*Cyberpunk Style*", "*Van Gogh*", "*Ukiyo-E*", "*Pixel Art*" and so on. The results are presented in Figure 11, which further enhances the diversity and complexity of $A^2$Bench.

Since most current animation methods Wang et al. (2024b); Hu et al. (2023); Zhang et al. (2024) take a pose image sequence as motion source, we also provide our $A^2$Bench with additional pose images. To achieve this, we employ DWPose Yang et al. (2023) to extract pose sequences from the videos. However, since DWPose is trained on human data, it does not accurately extract every pose in the dancing video of the anthropomorphic character, so after extraction, we manually screen 100 videos with accurate poses, and view them as test videos for calculating quantitative metrics. Fig. 3 displays several examples, which include anthropomorphic characters of plants, animals, food, furniture, etc. For images and videos where pose extraction is not feasible, we take them as key sources of reference images in our qualitative demonstrations. This will inspire the community to animate a wider range of interesting cases. We also anticipate that these data could serve as an important resource for future pose extraction algorithms tailored to anthropomorphic datasets, making them accessible for broader use.

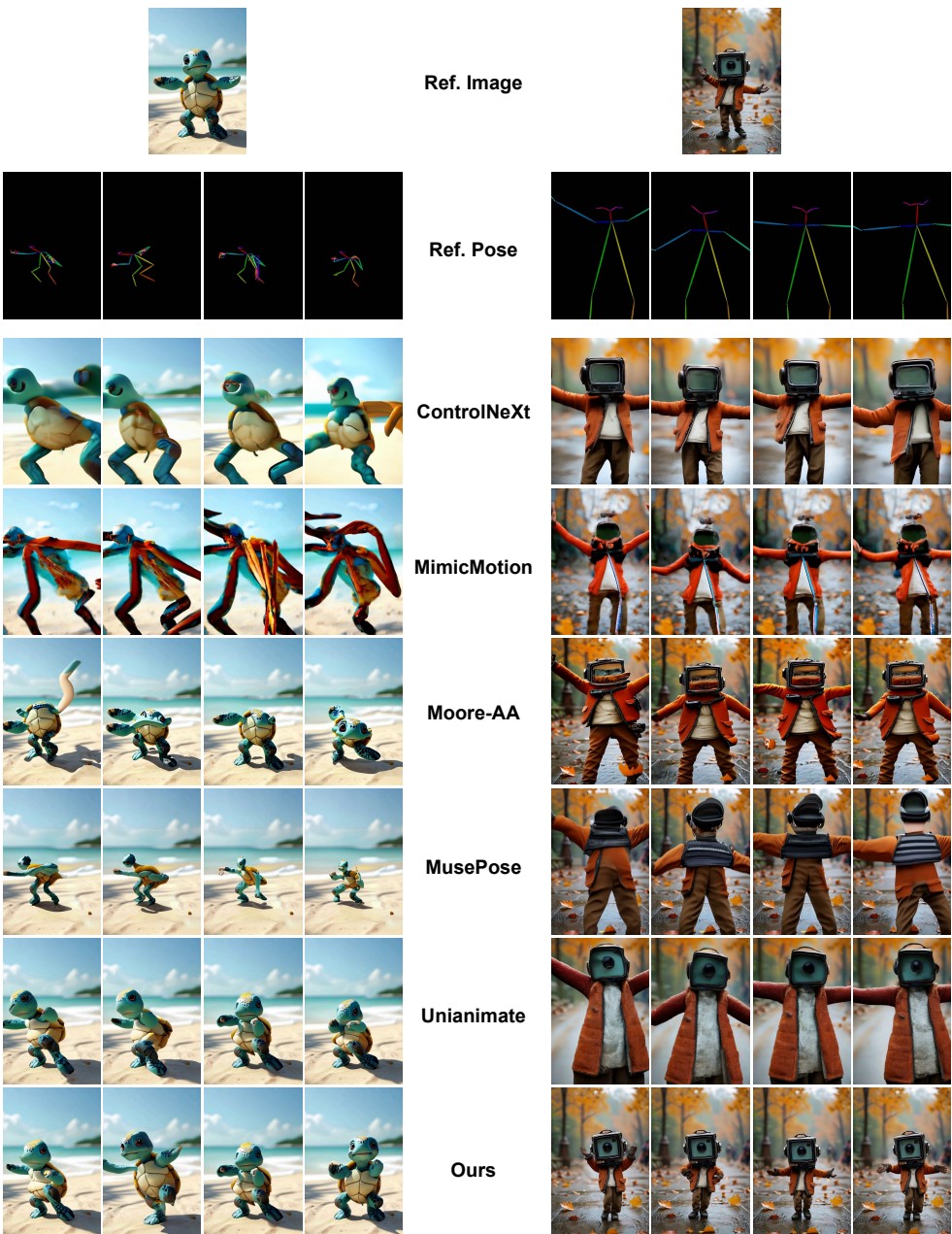

Figure 12: Visualization of cases in the user study

## C  USER STUDY

In Fig. 12, we present examples shown to participants for evaluation in our user study. To obtain genuine feedback reflective of practical applications, the ten participants in our user study experiment come from diverse academic backgrounds. Since many of them do not major in computer vision, we provide detailed explanations for each question to assist their judgments.

- Identity Preservation: By comparing the reference image with the two generated videos by different methods, determine which video's character more closely resembles the character in the image.

- Temporal Consistency: Evaluate the motion changes of the character within the video and compare which video exhibits more coherent movement.

- Visual Quality: Compared to the previous two questions, this one involves more subjective judgment. Participants should assess the videos comprehensively based on visual content (e.g., flashes, distortions, afterimages), motion effects (e.g., smoothness, physical logic), and overall plausibility.

## D ADDITIONAL EXPERIMENTAL RESULTS

### D.1 MORE QUALITATIVE RESULTS

In the main paper, we present qualitative comparison results between our method and the state-of-the-art (SOTA) methods under a cross-driven setting on a human-like character, where our approach demonstrates outstanding performance. Considering that the other methods are primarily self-driven and trained on human characters, making them more suitable for inference in such settings, we additionally provide comparison results under a self-reconstruction setting on Tiktok and Abench. As shown in Fig. 17, when there is a appreciably difference between the reference pose and the reference image, the GAN-based LIA Wang et al. (2022) produces noticeable artifacts. Thanks to the powerful generative capabilities of diffusion models, diffusion-based models generate higher-quality results. However, MusePose Tong et al. (2024) and MimicMotion Zhang et al. (2024) generate awkward arms and blurry hands, respectively, while ControlNeXt Peng et al. (2024) synthesizes incorrect movements. Only Unianimate Wang et al. (2024b) can obtain results comparable to ours. Yet, when the reference image is a non-human character, even in a self-driven setting with the same training strategy as Unianimate, their results still show distorted heads. Fig. 18 provides results of more comparison results, including MRAA Siarohin et al. (2021a), MagicAnimate Xu et al. (2023a) and Moore-AnimateAnyone Corporation (2024). In contrast, our method consistently generates satisfactory results for both human and anthropomorphic characters, demonstrating its ability to drive X character and highlighting its strong generalization and robustness.

### D.2 MORE QUANTITATIVE RESULTS

Tab. 10 and Tab. 11 presents the quantitative results on TikTok Jafarian & Park (2021) and Fashion Zablotskaia et al. (2019a) dataset, which suggests the superiority of methods over the comparison SOTA methods. Only Unianimate achieves comparable performance; however, our method is applicable to a wider range of characters and various unaligned pose inputs, as demonstrated in Tab. 1. This addresses the main issue that this paper aims to solve: developing a universal character image animation model.

### D.3 ROBUSTNESS

Our method demonstrates robustness to both input X character and pose variations. On the one hand, as shown in Fig. 1, our approach successfully handles inputs from diverse subjects, including characters vastly different from humans, such as those without limbs, as well as game characters or those generated by other models. Despite these variations, our method consistently produces satisfactory results without crashing, showcasing its robustness to the input reference images. On the other hand, as illustrated in Fig. 13, even when the pose images exhibit body part omissions (highlighted by the red circles), our method correctly interprets the intended motion and generates coherent results for the reference images. This highlights the robustness of our approach to different pose images.

### D.4 $A^2$BENCH

**Difficulty Level.** We add the difficulty level split for Animate-X. As shown in Figure 14, we categorize the videos in $A^2$Bench into three difficulty levels: Level 1, Level 2, and Level 3. The classification is based on their appearance characteristics. **First**, we classify characters that have body shapes and other appearance features similar to humans, as shown in the first row of Figure 14, into the easiest, Level 1 category. These characters are generally simpler to drive, produce fewer

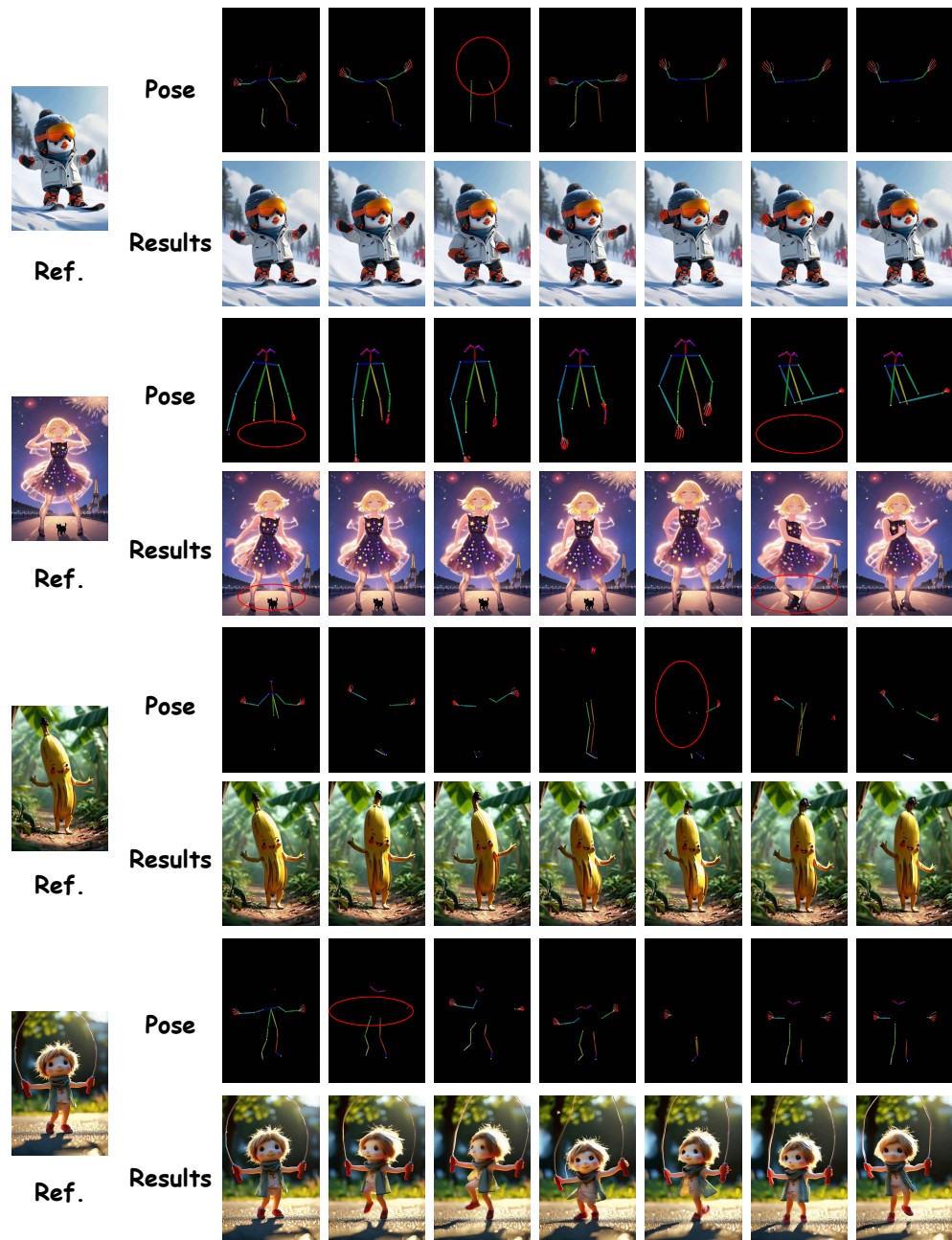

Figure 13: Visualization of the robustness of `Animate-X`.

artifacts, and have better motion consistency. **In contrast**, characters that maintain more distinct structural features from humans, such as dragons and ducks in the third row of Figure 14, are classified into the most difficult Level 3 category. These characters often preserve their original structures (*e.g.*, a duck's webbed feet and wings), which makes balancing identity preservation and motion consistency more challenging. To ensure identity preservation, the consistency of motion may be compromised, and vice versa. Additionally, images involving interactions between characters, objects, environments, and backgrounds are also placed in Level 3, as they increase the difficulty for the model to distinguish the parts that need to be driven from those that do not. **Videos in between these two categories**, like those in the second row of Figure 14, are classified as Level 2. These characters often strike a good balance between anthropomorphism and their original form, making

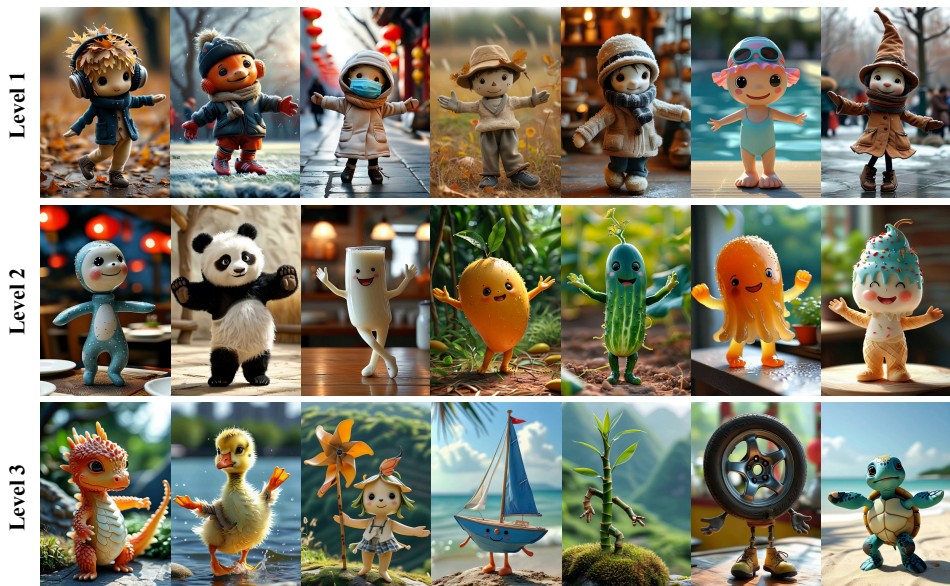

Figure 14: Difficulty levels in $A^2$`Bench`.

| Model-Level | PSNR* ↑ | SSIM ↑ | L1 ↓ | LPIPS ↓ | FID ↓ | FID-VID ↓ | FVD ↓ |
|---|---|---|---|---|---|---|---|
| `Animate-X`-level1 | 13.96 | 0.461 | 9.67E-05 | 0.418 | 24.24 | 31.37 | 681.53 |
| `Animate-X`-level2 | 13.74 | 0.457 | 9.82E-05 | 0.429 | 26.12 | 32.19 | 693.63 |
| `Animate-X`-level3 | 13.17 | 0.442 | 1.11E-04 | 0.437 | 27.34 | 35.64 | 721.41 |
| UniAnimate-level1 | 11.93 | 0.413 | 1.14E-04 | 0.521 | 42.39 | 52.14 | 1120.45 |
| UniAnimate-level2 | 11.89 | 0.408 | 1.20E-04 | 0.526 | 46.27 | 58.53 | 1147.34 |
| UniAnimate-level3 | 10.91 | 0.379 | 1.35E-04 | 0.549 | 56.58 | 65.39 | 1204.53 |

Table 5: User study results.

| Method | PSNR* ↑ | SSIM ↑ | L1 ↑ | LPIPS ↓ | FID ↓ | FID-VID ↓ | FVD ↓ |
|---|---|---|---|---|---|---|---|
| w/o IPI | 13.30 | 0.433 | 1.35E-04 | 0.454 | 32.56 | 64.31 | 893.31 |
| w/o LQ | 13.48 | 0.445 | 1.76E-04 | 0.454 | 28.24 | 42.74 | 754.37 |
| w/o DQ | 13.39 | 0.445 | **1.01E-04** | 0.456 | 30.33 | 62.34 | 913.33 |
| PA | 13.25 | 0.436 | 1.11E-04 | 0.464 | 27.63 | 46.54 | 785.36 |
| KV_Q | 13.34 | 0.443 | 1.17E-04 | 0.459 | 26.75 | 42.14 | 785.69 |
| w/o EPI | 12.63 | 0.403 | 1.80E-04 | 0.509 | 42.17 | 58.17 | 948.25 |
| w/o Add | 13.28 | 0.442 | 1.56E-04 | 0.459 | 34.24 | 52.94 | 804.37 |
| w/o Drop | 13.36 | 0.441 | 1.94E-04 | 0.458 | 26.65 | 44.55 | 764.52 |
| w/o BS | 13.27 | 0.443 | 1.08E-04 | 0.461 | 29.60 | 56.56 | 850.17 |
| w/o NF | 13.41 | 0.446 | 1.82E-04 | 0.455 | 29.21 | 56.48 | 878.11 |
| w/o AL | 13.04 | 0.429 | 1.04E-04 | 0.474 | 27.17 | 33.97 | 765.69 |
| w/o Rescalings | 13.23 | 0.438 | 1.21E-04 | 0.464 | 27.64 | 35.95 | 721.11 |
| w/o Realign | 12.27 | 0.433 | 1.17E-04 | 0.434 | 34.60 | 49.33 | 860.25 |
| **`Animate-X`** | **13.60** | **0.452** | 1.02E-04 | **0.430** | **26.11** | **32.23** | **703.87** |

Table 6: Quantitative results of ablation study.

them easier to animate with better motion consistency than Level 3 characters and more interesting results than Level 1 characters. We evaluate the results of `Animate-X` and UniAnimate for each subset. As shown in Tab. 5, as the difficulty increases, each evaluation result shows a decline.

**Motivation of T2I+I2V for $A^2$`Bench`.** The choice to use T2I models stems from a clear need: current T2V models often struggle with imaginative and logically complex inputs, such as "*personified refrigerators*" or "*human-like bees*". T2I models offer strict logic and imagination in these scenarios, allowing to generate reasonable cartoon characters as the ground-truth. To prove this point, as

shown in Table 7, we assess the semantic accuracy of $A^2$Bench using CLIP scores, which are commonly used to evaluate whether the semantic logic of images and text is strictly aligned (*i.e.*, Does the generated "*human-like bee*" maintain the visual essence of a bee while seamlessly incorporating human-like features, such as hands and feet?). We also add other metrics from VBench Huang et al. (2024), such as *Background Consistency*, *Motion Smoothness*, *Aesthetic Quality* and *Image Quality*, to assess the spatial and temporal consistency of the videos in $A^2$Bench. For comparison, we also evaluate the publicly available TikTok and Fashion datasets using the same metric. As shown in Table 7, $A^2$Bench achieves the highest level of strict logical alignment. $A^2$Bench outperforms TikTok dataset in all aspects and achieve comparable scores to Fashion dataset, where both TikTok and Fashion are collected from real-world scenarios. It demonstrates that the video generated by our method has the same level of spatial and temporal consistency as the real videos.

**Furthermore**, we input the images from $A^2$Bench into a multimodal large language model (MLLM) with logical reasoning, such as QWen Bai et al. (2023), to conduct a logical analysis of the visual outputs generated by the T2I model. The results, shown in Figure 15, reveal that the image descriptions answered by the MLLM closely aligns with our input prompts, which verifies again that the data in $A^2$Bench maintains strict logic.

Table 7: Quantitative results of different benchmarks. The best and second results for each column are **bold** and underlined, respectively.

| Benchmark | CLIP Score | Background Consistency | Motion Smoothness | Aesthetic Quality | Image Quality |
|---|---|---|---|---|---|
| TikTok | 26.92 | 94.10 % | 99.05 % | 55.14 % | 62.54 % |
| Fashion | 20.18 | **98.25** % | **99.45** % | 49.62 % | 49.96 % |
| $A^2$Bench | **33.24** | 96.66 % | 99.39 % | **69.86** % | **69.32** % |

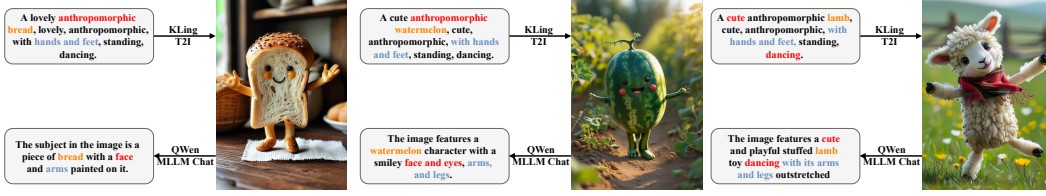

Figure 15: Prompts, generated images by T2I in $A^2$Bench, and logical answers from QWen.

## D.5 MORE ABLATION STUDY

In the main paper, we present the results of the primary ablation experiments for IPI and EPI. In this section, we supplement those results with additional ablation experiments to further demonstrate the contribution of each individual module.

**Ablation on Implicit Pose Indicator.** For more detailed analysis about the structure of IPI, we set up several variants: (1) remove IPI: w/o IPI. (2) remove learnable query: w/o LQ. (3) remove DWPose query: w/o DQ. (4) set IPI and spatial Attention to Parallel: PA. (5) set CLIP features as Q and DWPose as K,V in IPI: KV_Q. The quantitative results are shown in Tab. 6. It can be seen that removing the entire IPI presents the worst performance. By modifying the IPI module, although it improves on the w/o IPI, it still falls short of the final result of Animate-X, which suggests that our current IPI structure is the most reasonable and achieves the best performance.

Since IPI is embedded in Animate-X in the form of residual connection, i.e., $x = x + \alpha IPI(x)$, we also explore the impact of the weight $\alpha$ of IPI on performance as illustrated in Fig. 16, as $\alpha$ increases from 0 to 1, all metrics show a stable improvement despite some fluctuations. The best performance is achieved when $\alpha$ is set to 1, so we empirically set $\alpha$ to 1 in the final configuration.

**Ablation on Explicit Pose Indicator.** We conduct more detailed ablation experiments for different pairs of pose transformations by (1) removing the entire EPI: w/o EPI; (2)&(3) removing adding and dropping parts; canceling the change of the length of (4) body and should: w/o BS; (5) neck and face: w/o NF; (6) arm and leg: w/o AL; (7) removing all rescaling process: w/o Rescalings; (8)

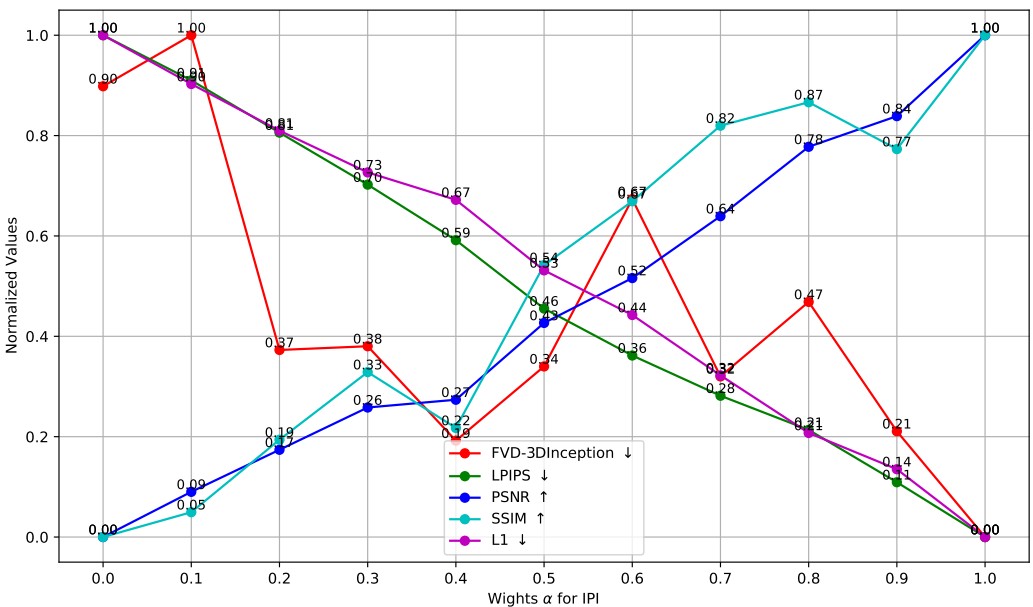

Figure 16: Ablation study on the weight $\alpha$ of Implicit Pose Indicator. To better visualize the impact of $\alpha$ on performance, we normalize all the values to the range of 0 to 1.

remove another person pose alignment: w/o Realign. From the results displayed in Tab. 6, we found that each pose transformation contributes compared to w/o EPI, with aligned transformations with another person's pose contributing the most. It suggests that maintaining the overall integrity of the pose while allowing for some variations is the most important factor, and EPI also learns the overall integrity of the pose. The final result indicates that all the transformations together achieve the best performance.

To explore the effect of different probabilities $\lambda$ of using pose transformation for EPI on the model performance, we set $\lambda$ as 100%, 98%, 95%, 90% and 80% for the ablation experiments on two datasets. The results presented in Tab. 9 suggest that a high $\lambda$ performs better on $A^2$Bench, i.e., it performs better when the reference image and pose image are not aligned, but harms performance on the TikTok dataset, i.e., when the reference image and pose image are strictly aligned. In contrast, a relatively low $\lambda$, e.g., 90%, would be in this case perform better. It is reasonable that in the case of strict alignment, we expect the pose to provide a strictly accurate motion source, and thus need to reduce the percentage $\lambda$ of pose transformation. However, in the non-strictly aligned case, we expect the pose image to provide an approximate motion trend, so we need to increase $\lambda$.

Since the anchor poses are chosen from the entire training set, we further conduct the statistical analysis for rescaling ratio. First, we randomly sample a driven pose $I^p$ and then traverse the entire pose pool, treating each pose in the pool as an anchor pose to calculate the rescaling ratio. We repeat this process 10 times. Finally, we divide the range from 0.001 to 10 into 10 intervals, counting the proportion of rescaling ratios that fell within each interval. We analyze the proportions of other important parts like shoulder length, body length, upper arm length, lower arm length, upper leg length, lower leg length. As shown in Tab. 8, the overall distribution covers a wide range (from 0.001 to 10.0), which allows the model to learn poses of various characters, encompassing non-human subjects.

# E   DISCUSSION

## E.1   LIMITATION AND FUTURE WORK

Although our method has made remarkable progress, it still has certain limitations. Firstly, its ability to model hands and faces remains insufficient, a limitation commonly faced by most current

| Interval | Shoulder Length | Body Length | Upper Arm Length | Lower Arm Length | Upper Leg Length | Lower Leg Length |
|---|---|---|---|---|---|---|
| [0.001, 0.1) | 0.19% | 0.14% | 0.05% | 0.08% | 0.05% | 0.81% |
| [0.1, 0.3) | 1.52% | 5.73% | 4.04% | 3.22% | 0.59% | 4.60% |
| [0.3, 0.5) | 12.21% | 18.57% | 15.28% | 7.63% | 4.26% | 5.65% |
| [0.5, 0.7) | 15.33% | 16.93% | 12.97% | 7.54% | 12.02% | 9.61% |
| [0.7, 1.0) | 20.07% | 18.48% | 17.15% | 11.35% | 24.86% | 19.53% |
| [1.0, 1.5) | 22.09% | 18.63% | 17.56% | 15.38% | 27.90% | 24.89% |
| [1.5, 2) | 10.07% | 8.34% | 7.93% | 11.73% | 14.31% | 14.47% |
| [2.0, 3.0) | 9.75% | 6.52% | 7.73% | 16.19% | 11.83% | 15.28% |
| [3.0, 6.0) | 6.33% | 6.28% | 10.93% | 18.40% | 2.73% | 4.30% |
| [6.0, 10.0) | 2.43% | 0.37% | 6.37% | 8.47% | 1.45% | 0.85% |

Table 8: Statistical analysis for rescaling ratio.

| Method | $A^2$Bench | | | | TikTok Jafarian & Park (2021) | | | |
|---|---|---|---|---|---|---|---|---|
| | SSIM↑ | FID↓ | FID-VID↓ | FVD↓ | SSIM↑ | FID↓ | FID-VID↓ | FVD↓ |
| 100% | **0.452** | **26.11** | **32.23** | **703.87** | 0.802 | 55.26 | 17.47 | 138.36 |
| 98% | 0.448 | 26.93 | 37.67 | 775.24 | 0.797 | 55.81 | 16.28 | 129.48 |
| 95% | 0.447 | 27.46 | 39.21 | 785.55 | 0.804 | **52.72** | 14.61 | **124.92** |
| 90% | 0.444 | 27.15 | 38.03 | 775.38 | **0.806** | 52.81 | 14.82 | 139.01 |
| 80% | 0.442 | 29.13 | 47.93 | 803.97 | 0.802 | 54.51 | **14.42** | 133.78 |

Table 9: Quantitative results for different probabilities of using pose transformation.

| Method | L1 ↓ | PSNR ↑ | PSNR* ↑ | SSIM ↑ | LPIPS ↓ | FVD ↓ |
|---|---|---|---|---|---|---|
| FOMM Siarohin et al. (2019a) (NeurIPS19) | 3.61E-04 | - | 17.26 | 0.648 | 0.335 | 405.22 |
| MRAA Siarohin et al. (2021a) (CVPR21) | 3.21E-04 | - | 18.14 | 0.672 | 0.296 | 284.82 |
| TPS Zhao & Zhang (2022a) (CVPR22) | 3.23E-04 | - | 18.32 | 0.673 | 0.299 | 306.17 |
| DreamPose Karras et al. (2023) (ICCV23) | 6.88E-04 | 28.11 | 12.82 | 0.511 | 0.442 | 551.02 |
| DisCo Wang et al. (2024a) (CVPR24) | 3.78E-04 | 29.03 | 16.55 | 0.668 | 0.292 | 292.80 |
| MagicAnimate Xu et al. (2023a) (CVPR24) | 3.13E-04 | 29.16 | - | 0.714 | 0.239 | 179.07 |
| Animate Anyone Hu et al. (2023) (CVPR24) | - | 29.56 | - | 0.718 | 0.285 | 171.90 |
| Champ Zhu et al. (2024) (ECCV24) | 2.94E-04 | 29.91 | - | 0.802 | 0.234 | 160.82 |
| Unianimate Wang et al. (2024b) (ArXiv24) | **2.66E-04** | 30.77 | 20.58 | **0.811** | **0.231** | 148.06 |
| MusePose Tong et al. (2024) (ArXiv24) | 3.86E-04 | - | 17.67 | 0.744 | 0.297 | 215.72 |
| MimicMotion Zhang et al. (2024) (ArXiv24) | 5.85E-04 | - | 14.44 | 0.601 | 0.414 | 232.95 |
| ControlNeXt Peng et al. (2024) (ArXiv24) | 6.20E-04 | - | 13.83 | 0.615 | 0.416 | 326.57 |
| **Animate-X** | 2.70E-04 | **30.78** | **20.77** | 0.806 | 0.232 | **139.01** |

Table 10: Quantitative comparisons with existing methods on TikTok dataset.

| Method | PSNR ↑ | PSNR* ↑ | SSIM ↑ | LPIPS ↓ | FVD ↓ |
|---|---|---|---|---|---|
| MRAA Siarohin et al. (2021a) (CVPR21) | - | - | 0.749 | 0.212 | 253.6 |
| TPS Zhao & Zhang (2022a) (CVPR22) | - | - | 0.746 | 0.213 | 247.5 |
| DPTN Zhang et al. (2022a) (CVPR22) | - | 24.00 | 0.907 | 0.060 | 215.1 |
| NTED Ren et al. (2022) (CVPR22) | - | 22.03 | 0.890 | 0.073 | 278.9 |
| PIDM Bhunia et al. (2023) (CVPR23) | - | - | 0.713 | 0.288 | 1197.4 |
| DBMM Yu et al. (2023) (ICCV23) | - | 24.07 | 0.918 | 0.048 | 168.3 |
| DreamPose Karras et al. (2023) (ICCV23) | - | - | 0.885 | 0.068 | 238.7 |
| DreamPose w/o Finetune Karras et al. (2023) (ICCV23) | 34.75 | - | 0.879 | 0.111 | 279.6 |
| Animate Anyone Hu et al. (2023) (CVPR24) | **38.49** | - | 0.931 | 0.044 | 81.6 |
| Unianimate Wang et al. (2024b) (ArXiv24) | 37.92 | 27.56 | **0.940** | **0.031** | **68.1** |
| MimicMotion Zhang et al. (2024) (ArXiv24) | - | 27.06 | 0.928 | 0.036 | 118.48 |
| **Animate-X** | 36.73 | **27.78** | **0.940** | **0.030** | 79.4 |

Table 11: Quantitative comparisons with existing methods on the Fashion dataset. "w/o Finetune" represents the method without additional finetuning on the fashion dataset.

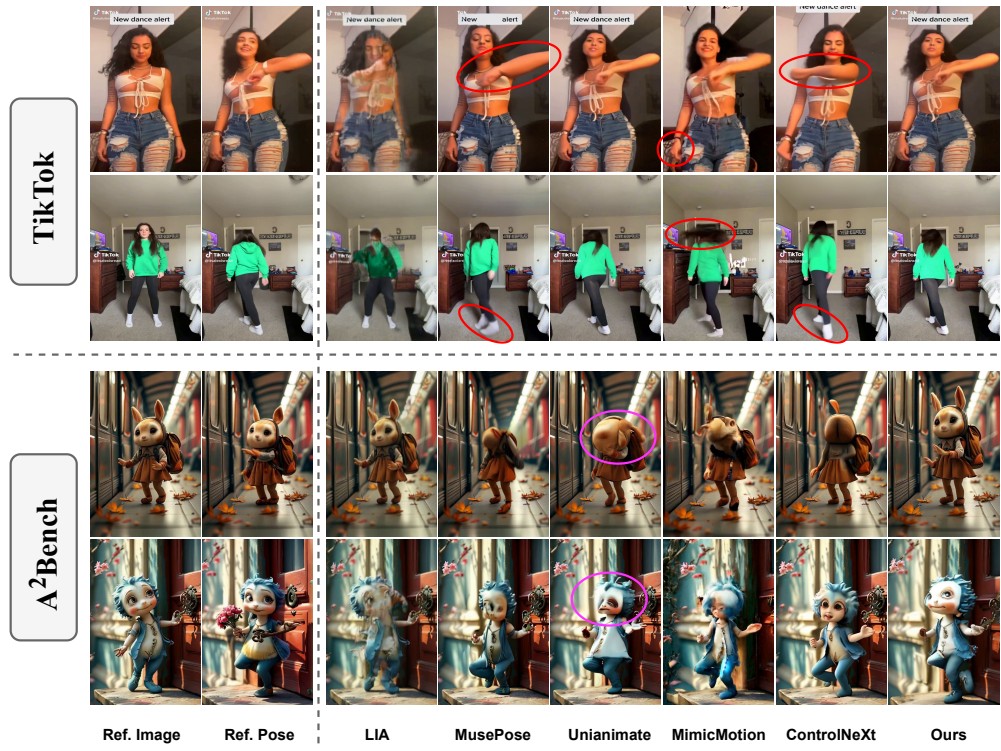

Figure 17: Visualization comparison on TikTok dataset and $A^2$Bench.

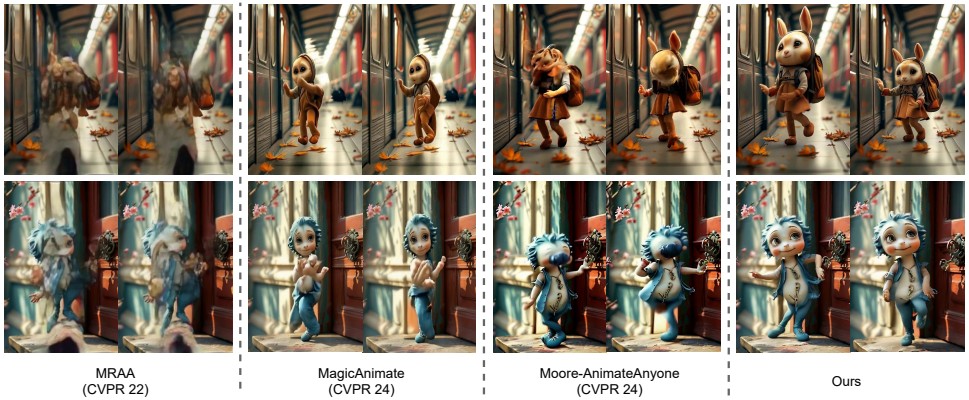

Figure 18: Comparison with more SOTAs on $A^2$Bench.

generative models. While our **IPI** leverages CLIP features to extract implicit information such as motion patterns from the driving video, mitigating the reliance on potentially inaccurate hand and face detection by DWPose, there is still a gap between our results and the desired realism. Secondly, due to the multiple denoising steps in the diffusion process, even though we replace the transformer with a more efficient Mamba model for temporal modeling, Animate-X still cannot achieve real-time animation. In future work, we aim to address these two limitations. Additionally, we will focus on studying interactions between the character and the surrounding environment, such as the background, as a key task to resolve. As for $A^2$Bench, creating 3D models and rendering them with predefined actions using tools like Blender and Maya is a superior approach for developing a character benchmark, which is also part of our future work.

### E.2 ETHICAL CONSIDERATIONS

Our approach focuses on generating high-quality character animation videos, which can be applied in diverse fields such as gaming, virtual reality, and cinematic production. By providing body movement, our method enables animators to create more lifelike and dynamic characters. However, the potential misuse of this technology, particularly in creating misleading or harmful content on digital platforms, is a concern. While greatly progress has been made in detecting manipulated animations Boulkenafet et al. (2015); Wang et al. (2020); Yu et al. (2020), challenges remain in accurately identifying increasingly sophisticated forgeries. We believe that our animation results can contribute to the development of better detection techniques, ensuring the responsible use of animation technology across different domains.

