# OpenReview forum: "Animate-X: Universal Character Image Animation with Enhanced Motion Representation"
_ICLR.cc/2025/Conference — ICLR 2025 Poster_

### Official Review · Reviewer_ESK8 · 2024-10-31

**Soundness:** 3
**Presentation:** 3
**Contribution:** 3
**Rating:** 6
**Confidence:** 5

**Summary:**

This paper proposed Animate-X, a universal animation framework based on diffusion models. The key insight of this work is that existing image animation frameworks are only focused on the human images and fail to extract the movement pattern of the driving video, leading to the rigid retargeting of the driving pose to the target reference image. The authors propose two pose indicators to address this issue, which can capture comprehensive motion patterns from the driving video. The implicit pose indicator helps retrieve relevant features from the driving video, while the explicit one simulates the unaligned driving poses in the inference stage. To evaluate the approaches, the authors also propose a new benchmark which contains in-the-wild and unmatched driving sequence/reference image pairs. Experiments show that the proposed method outperforms state-of-the-art methods.

**Strengths:**

1. The motivation of this work is clear, it comes from an in-depth analysis of the failure cases of existing works. The alignment between the driving signal and reference image is critical to the fidelity of character animation. The authors propose an effective method to tackle this problem.
2. The experimental results, especially the video results, are reasonable and interesting. The proposed method shows state-of-the-art performance and outperforms baselines in animating in-the-wild characters. This indicates that the training data is well utilised and shows that the proposed method helps improve the generalisation ability of the animation model.
3. The evaluation benchmark is valuable to the research community. It can help follow-up works measure their methods comprehensively.

**Weaknesses:**

1. The backbone of this work remains unchanged, it is quite similar to the prior works like your reference AnimateAnyone and MagicAnimate, which makes this work a straightforward extension of existing works and thus reduces the contribution of this paper.
2. Leveraging driving videos to boost the animation performance has been already explored in a few prior works like [1]. The implicit pose indicator is also a similar design which aims to extract comprehensive motion patterns to improve the animation performance.
3. The explicit pose indicator is a little bit confusing because I think this module is an augmentation of the driving pose sequences. Therefore, the novelty of the proposed method is not very significant. It is reasonable that the augmentation can break the strong correspondence between the driving video and motion representation. What is the advantage of this training time rescale augmentation and over the test time pose alignment? Are there any ablation studies about this?
4. From the results of the animation of anthropomorphic characters, the example of a banana shows that although the animation result looks like a banana, the motion precision is decreased. Therefore, I think the implicit pose indicator could harm the motion precision. The authors could conduct more experiments to study this issue.

[1] X-portrait: Expressive portrait animation with hierarchical motion attention

**Questions:**

Does this model still use any input videos in the inference stage? I am asking this question because there are no input driving videos in the “Animating anthropomorphic characters” section of the supplementary materials. Could the author explain the inference setting? If there is a corresponding driving video, it is better to also include them into the results.

---

> ### Author Response · Authors · 2024-11-20
> **Thank you for your valuable feedbacks.**
>
> We sincerely thank **Reviewer #4 ESK8** for acknowledging *"the clear motivation, video results, and benchmark presented in our work"*. We have re-uploaded our supplementary materials, which include the complete responses (at `.zip/Animate-X_rebuttal_response_letter.pdf`) along with the relevant figures and tables. The response letter is also contained in the main paper, after page 25. Below, we have addressed each question in detail and hope to clarify any concerns.
>
> **Comment 1: The backbone of this work remains unchanged, it is quite similar to the prior works like your reference AnimateAnyone and MagicAnimate, which makes this work a straightforward extension of existing works and thus reduces the contribution of this paper.**
>
> Thanks for the comments. First of all, the primary contribution of this work is the introduction of the **universal** character image animation. We proposed Animate-X to addresses challenges by leveraging our proposed IPI and EPI modules to implicitly and explicitly model the universal pose indicator. Using the same backbone as AnimateAnyone and MagicAnimate, which have pioneered in latent diffusion models for human animation, allows us to have a fair comparison with these works and demonstrate the contribution of IPI and EPI to animate anthropomorphic figures.
>
>
> ***
>
> **Comment 2: Leveraging driving videos to boost the animation performance has been already explored in a few prior works like [1]. The implicit pose indicator is also a similar design which aims to extract comprehensive motion patterns to improve the animation performance. [1] X-portrait: Expressive portrait animation with hierarchical motion attention.**
>
> Thanks for the comments and for introducing X-Portrait. We will cite it and discuss the difference between X-Portrait and ours:
> - **1. Use of the Driven Video:** In Animate-X, we extract pose images from the driven video to serve as the primary source of motion. Given that a single pose image cannot provide image-level motion-related details (such as motion-induced deformations like body part overlap, occlusion, and overall motion patterns). In contrast, X-Portrait directly inputs the driven video into the model without any processing, which is following most of GAN-based animation methods.
> - **2. Different Technical Approaches:** X-Portrait follows the approach of ControlNet, where the driving video is fed into an SD U-Net, and then a zero-conv layer is inserted into the main branch of the U-Net. In comparison, our IPI module first uses a pre-trained CLIP encoder to extract features from the driven video and then decouples image-level motion-related features for motion modeling.
> - **3. Task Scope:** X-Portrait focuses on facial animation, but Animate-X handles full-body animation for universal characters, which includes anthropomorphic figures in cartoons and games.
>
> In summary, Animate-X is different from X-Portrait in *Use of the Driven Video*, *Technical Approaches*, and *Task Scope*.

---

> ### Author Response · Authors · 2024-11-20
> **Response (2)**
>
> **Comment 3: The explicit pose indicator is a little bit confusing because I think this module is an augmentation of the driving pose sequences. Therefore, the novelty of the proposed method is not very significant. It is reasonable that the augmentation can break the strong correspondence between the driving video and motion representation. What is the advantage of this training time rescale augmentation and over the test time pose alignment (Answer 3.1)? Are there any ablation studies about this? (Answer 3.2)**
>
> **Answer 3.1:** The advantages of training time rescale augmentation over the test time alignment are as follows:
> - **1. Generalization for Characters Without Extractable Poses:** For reference images with structures significantly different from human skeletons, such as the limb-less fairy shown in Figure 1 (original submission), pose extraction using DWPose is not feasible, which is because DWPose is specifically designed for processing human poses. Consequently, pose alignment at test time cannot be performed, making the diffusion model challenging to generate reasonable videos. In contrast, training time rescale augmentation enables the diffusion model to learn how to handle misaligned reference and driven poses, enhancing its robustness and generalization. In this way, Animate-X can handle scenarios where poses cannot be extracted from the reference image, as it eliminates the need for pose alignment between the reference and driven pose images during inference.
> - **2. Reduced Dependency on Strict Pose Alignment:** Even when pose alignment is available at test time, the results often rely heavily on precise alignment. For example, if the aligned pose differs in arm length from the reference image (*e.g.*, a longer arm), the generated result will reflect this discrepancy, compromising identity preservation. In contrast, rescale augmentation during training reduces the model’s dependence on strict pose alignment, ensuring that even with imperfect or absent alignment, the generated results can still effectively preserve identity information.
> - **3. Simpler Test-Time Workflow and Faster Inference:** For example, animating 100,000,000 reference images with a single driven pose using previous methods would require extracting the pose for each of the 100,000,000 reference images, followed by an equal number of strict pose alignment operations. In contrast, our method removes the need for these alignment operations, significantly reducing inference time and simplifying the test-time process.
>
> **Answer 3.2:** We have conducted extensive ablation experiments for different pairs of pose transformations in EPI, as detailed in Appendix D.4 and **Table X**. The results show that each pose transformation improves performance compared to the scenarios without augmentation, confirming the effectiveness of the augmentation operation in enhancing the model's performance.
>
> | **Method**                | **PSNR*** ↑   | **SSIM** ↑   | **L1** ↓         | **LPIPS** ↓   | **FID** ↓       | **FID-VID** ↓  | **FVD** ↓       |
> |---------------------------|---------------|--------------|------------------|---------------|-----------------|----------------|-----------------|
> | w/o Add in EPI            | 13.28         | 0.442        | 1.56E-04         | 0.459         | 34.24           | 52.94          | 804.37          |
> | w/o Drop in EPI           | 13.36         | 0.441        | 1.94E-04         | 0.458         | *26.65*         | 44.55          | 764.52          |
> | w/o BS in EPI             | 13.27         | 0.443        | 1.08E-04         | 0.461         | 29.60           | 56.56          | 850.17          |
> | w/o NF in EPI             | *13.41*       | *0.446*      | 1.82E-04         | 0.455         | 29.21           | 56.48          | 878.11          |
> | w/o AL in EPI             | 13.04         | 0.429        | *1.04E-04*       | 0.474         | 27.17           | *33.97*        | 765.69          |
> | w/o Rescalings in EPI     | 13.23         | 0.438        | 1.21E-04         | 0.464         | 27.64           | 35.95          | *721.11*        |
> | w/o Realign in EPI        | 12.27         | 0.433        | 1.17E-04         | *0.434*       | 34.60           | 49.33          | 860.25          |
> | w/o EPI                   | 12.63         | 0.403        | 1.80E-04         | 0.509         | 42.17           | 58.17          | 948.25          |
> | **Animate-X**             | **13.60**     | **0.452**    | **1.02E-04**     | **0.430**     | **26.11**       | **32.23**      | **703.87**      |
>
> **Table X:** Quantitative results of the ablation study. The best and second-best results for each metric are **bold** and *italicized*, respectively.

---

> ### Author Response · Authors · 2024-11-20
> **Response (3)**
>
> **Comment 4: From the results of the animation of anthropomorphic characters, the example of a banana shows that although the animation result looks like a banana, the motion precision is decreased. Therefore, I think the implicit pose indicator could harm the motion precision (Answer 4.1). The authors could conduct more experiments to study this issue (Answer 4.2).**
>
> **Answer 4.1:** First of all, we need to clarify that the implicit pose indicator does not harm motion precision. We have demonstrated that adding the IPI module to the baseline results in improvements across all quantitative metrics, highlighting its contributions to every aspect of animation through extensive ablation experiments (i.e., **Table III**).
>
> | **Method**        | **PSNR*** ↑   | **SSIM** ↑     | **L1** ↓         | **LPIPS** ↓     | **FID** ↓       | **FID-VID** ↓  | **FVD** ↓       |
> |--------------------|---------------|----------------|------------------|-----------------|-----------------|----------------|-----------------|
> | w/o IPI           | 13.30         | 0.433          | 1.35E-04         | *0.454*         | 32.56           | 64.31          | 893.31          |
> | w/o LQ            | *13.48*       | 0.445          | 1.76E-04         | *0.454*         | 28.24           | 42.74          | 754.37          |
> | w/o DQ            | 13.39         | 0.445          | **1.01E-04**     | 0.456           | 30.33           | 62.34          | 913.33          |
> | **Animate-X**      | **13.60**     | **0.452**      | *1.02E-04*       | **0.430**       | **26.11**       | **32.23**      | **703.87**      |
>
> **Table III:** Quantitative results of the ablation study on IPI. The best and second-best results for each column are **bold** and *italicized*, respectively.
>
> **Answer 4.2:** As shown in Figure 12 (response letter), we have conducted additional experiments on the banana case and provided a detailed discussion. Specifically, we input the banana image and the driven poses into the model without the IPI module to generate the results. As shown in Figure 12 (response letter), we observe that without the IPI module, the model generates the human-like arms, which was not the intended outcome. In contrast, Animate-X (with IPI) prioritized preserving the banana's identity and avoiding obvious artifacts. We believe this trade-off is reasonable and aligns with the limitation discussed in our paper: the excessive sacrifices in identity preservation in favor of strict pose consistency.
>
> To balance pose consistency and identity preservation, we assigned an appropriate weight to the IPI module. In this way, we generated the preferrable result, as shown in the last row of Figure 12 (response letter). To allow users to control the trade-off, we made this weight an adjustable parameter. Additionally, we conducted detailed experiments and analysis of this weight, as presented in Figure 12 (original submission).
>
> ***
>
> **Comment 5: Does this model still use any input videos in the inference stage (Answer 5.1)? I am asking this question because there are no input driving videos in the “Animating anthropomorphic characters” section of the supplementary materials. Could the author explain the inference setting  (Answer 5.2)? If there is a corresponding driving video, it is better to also include them into the results (Answer 5.3).**
>
> **Answer 5.1:** Yes, this model can still use any input videos during the inference stage.
>
> **Answer 5.2:** Yes. As shown in Figure 13 (response letter), during inference, our method takes a reference image and a driven video as input and outputs an animated video that maintains the same identity as the reference image and the same motion as the driven video.
>
> **Answer 5.3:** Thanks. Following your suggestions, we have included the corresponding driving video in the results. Please see the videos in (`.zip/for_reviewer_ESK8/for_comment_5/xxx.mp4`).

---

> ### Author Response · Authors · 2024-11-25
>
> Dear Reviewer ESK8,
>
> Thank you again for the great efforts and valuable comments. We hope you find the response satisfactory. As the discussion phase is about to close, we are eagerly looking forward to hearing from you regarding any further feedback. We will be more than happy to address any additional concerns you may have.
>
> Best,
>
> Animate-X Authors

---

> > ### Comment · Reviewer_ESK8 · 2024-11-27
> >
> > Dear authors,
> >
> > Thanks for your effort in the rebuttal. I have carefully read all the responses, sup mat, as well as the comments and responses of other reviewers. Most of my questions have been addressed by the rebuttal, and I choose to remain with the current rating. The reason for not raising is that this work is an extension of the prior works in this field. Considering the standard of this conference, its contributions are not significant enough.

---

> > > ### Author Response · Authors · 2024-11-27
> > >
> > > Dear Reviewer  ESK8,
> > >
> > > Thank you for dedicating your valuable time to review our work and for carefully reading our responses, supplementary materials, as well as the comments and responses of other reviewers.
> > >
> > > We are pleased to know that our responses have addressed your questions.
> > >
> > > If you have any further questions or concerns, please do not hesitate to reach out to us for further discussion.
> > >
> > > Best regards,
> > >
> > > Animate-X authors

---

### Official Review · Reviewer_feUz · 2024-11-03

**Soundness:** 3
**Presentation:** 3
**Contribution:** 3
**Rating:** 8
**Confidence:** 4

**Summary:**

This paper highlights that character animation models trained exclusively on human-only datasets struggle to learn motion patterns from driving videos, often leading to overfitting on the driving pose and poor generalization to anthropomorphic characters.

To address this issue, the authors propose a novel character animation framework called Animate-X, which incorporates two Pose Indicators. The Implicit Pose Indicator extracts motion and integrates it with CLIP features, while the Explicit Pose Indicator supports an augmentation pipeline during training that encourages the model to learn motion from misaligned pose sequences.

Additionally, a new benchmark is established for evaluating anthropomorphic characters. Experiments across multiple datasets demonstrate the effectiveness of the proposed method for animating anthropomorphic characters.

**Strengths:**

- The paper introduces a new augmentation method that enhances pose robustness for character animation techniques.

- A novel module is proposed to integrate the driving pose with the reference image without relying on a reference network.

- A new benchmark is established for evaluating anthropomorphic characters.

- The quality of animation results is good, even reference characters do not have leg or arm.

**Weaknesses:**

- The paper lacks a detailed analysis of the construction of the augmentation pool, making it difficult to reproduce the method.

- There is insufficient in-depth analysis of the model design, such as why the Implicit Pose Indicator (IPI) outperforms the reference network, which has more learnable parameters.

- Most styles in the A2Bench benchmark are "3D render style"; the benchmark should include a wider variety of visual styles.

**Questions:**

- Could the authors provide more details on the construction of the pose pool and alignment pool, such as the pool sizes and how poses are selected from the training set?

- Comparing the results in Table 4 and Table 1, Animate-X outperforms the baselines even without pose augmentation (EPI). Could the authors provide a deeper analysis of why the Implicit Pose Indicator (IPI), with fewer parameters, outperforms the reference network?

- What happens if the reference pose differs significantly from the candidates in the pose pool and alignment pool? The authors should provide a robustness analysis for this scenario and consider adding a difficulty level split for A2Bench.

- Could aligning the driving pose to a "standard" one in the pose pool further improve generation quality?

- In the supplementary materials, the authors show results in various styles, yet most styles in A2Bench are in "3D render style." Would it be possible to add a "style trigger word" in the prompt template to diversify and strengthen the benchmark?

---

> ### Author Response · Authors · 2024-11-20
> **Thank you for your valuable feedbacks.**
>
> We sincerely thank **Reviewer #3 feUz** for acknowledging *"the new method, benchmark, and animation results presented in our work"*.  We have re-uploaded our supplementary materials, which include the complete responses (at `.zip/Animate-X_rebuttal_response_letter.pdf`) along with the relevant figures and tables. The response letter is also contained in the main paper, after page 25. Below, we have addressed each question in detail and hope to clarify any concerns.
>
> **Comment 1: The paper lacks a detailed analysis of the construction of the augmentation pool, making it difficult to reproduce the method. Could the authors provide more details on the construction of the pose pool and alignment pool, such as the pool sizes and how poses are selected from the training set?**
>
> Thanks for your feedback. Yes, we present the detailed analysis of the construction of the augmentation pool. Please refer to Figure 4 (response letter) or Figure 8 (original submission) for an illustration of the following process:
> - **Step1:** We first construct the pose pool using the DWPose extractor. The pose pool is composed of pose skeletons (*i.e.*, pose images);
> - **Step2:** Given a driving pose $I^p$, we randomly select an anchor pose $I^p_{anchor}$ from the pose pool.
> - **Step3:** We then calculate the proportion of each body part between these two poses. For example, the shoulder length of $I^p_{anchor}$ divided by the shoulder length of $I^p$ might be 0.45, and the leg length of $I^p_{anchor}$ divided by the leg length of $I^p$ might be 0.53, and so on.
> - **Step4:** We multiply each body part of the driven pose (*i.e.*, $I^p$) by the corresponding ratio (*e.g.*, 0.45, 0.53, *etc.*) to obtain the aligned pose (*i.e.*, $I^p_n$).
> - **Step5:** Then we define a set of keypoint rescaling operations, including modifying the length of the body, legs, arms, neck, and shoulders, altering face size, adding or removing specific body parts, *etc*. These transformations are stored in a rescale pool.
> - **Step6:** We apply the selected transformations on the aligned pose $I^p_{realign}$ to obtain the final transformed poses $I^p_n$.

---

> > ### Comment · Reviewer_feUz · 2024-11-28
> > **Further question about selection of $I_{\mathcal{anchor}}^{p}$**
> >
> > Author(s) describe how transformed poses $I_n^p$ is generated during training. But I still have some corcerns regarding **how anchor poses** are selected? Specifically, are the anchor poses chosen from the entire training set or from a subset?
> >
> > * If they are randomlly selected from the entire training set, how does the static distribution of rescaling ratio (e.g., the shoulder length of $I_{\mathcal{anchor}}^p$ divided by the shoulder length of $I^p$) look like?
> > * If they are selected from a subset, what is the number of anchor poses, and what is the selection rule?

---

> ### Author Response · Authors · 2024-11-20
> **Response (2)**
>
> **Comment 2: Here is insufficient in-depth analysis of the model design, such as why the Implicit Pose Indicator (IPI) outperforms the reference network, which has more learnable parameters. Comparing the results in Table 4 and Table 1, Animate-X outperforms the baselines even without pose augmentation (EPI). Could the authors provide a deeper analysis of why the Implicit Pose Indicator (IPI), with fewer parameters, outperforms the reference network?**
>
> Sure, IPI outperforms the reference network because the latter focuses on extracting content features from reference images, while IPI focuses on motion, aiming to capture a universal motion representation. The reference network intends to capture all appearance details of the reference image. In contrast, IPI only models the motion-related image-level detais, so IPI can employ a smaller network to do the job. We provide a detailed explanation of how IPI improves the performance as follows:
> - **1. Reference network:** From the results using current methods using the reference network, *e.g.*, MimicMotion, we observe an inherent trade-off between overly precise poses and low fidelity to reference images. While the reference network attempts to address this by extracting additional appearance information from the reference image to improve fidelity through the denoising model, Figure 9 (response letter) illustrates that the reference network based approach remains insufficient, as precise human poses still dominate.
> - **2. IPI:** To address the observed limitations, we shifted our focus from appearance information to motion as the critical factor in our work. Simple 2D pose skeletons, constructed by connecting sparse keypoints, lack the image-level details needed to capture the essence of the reference video, such as motion-induced deformations (*e.g.*, body part overlap and occlusion). This absence of image-level details causes previous methods, even those using a reference network, to produce results with consistent poses but compromised identity fidelity. To overcome this issue, we introduced the IPI module to recover these missing **motion-related** image-level details. Specifically, IPI employs a pretrained CLIP encoder to extract features from the driving image, followed by a lightweight extractor ($P$) to isolate the motion-related details. This approach enables IPI to outperform the reference network, which, despite having more learnable parameters, unable to capture these essential motion-related features.
>
> As shown in Figure 9 (response letter), methods utilizing reference networks, such as AnimateAnyone, primarily focus on preserving colors from the reference image, as demonstrated by the white hat and yellow body of the potato in the first row. However, these methods cannot maintain the identity of the reference image, often generating videos that deviate from the original image, such as forcefully inserting human limbs onto potatoes. It highlights the limitation of reference networks, which prioritize color consistency over identity preservation, leading to weaker performance on quantitative metrics like SSIM, L1, and FID.
>
> In contrast, as shown in Figure 7 (original submission), even without the EPI module, Animate-X successfully generates a panda that retains the identity of the reference image. This leads to substantial improvements in SSIM, L1, and FID compared to baselines that rely on reference networks, even without the EPI module.

---

> ### Author Response · Authors · 2024-11-20
> **Response (3)**
>
> **Comment 3: What happens if the reference pose differs significantly from the candidates in the pose pool and alignment pool? The authors should provide a robustness analysis for this scenario.**
>
> Thanks. We are a bit unsure whether the reviewer's question refers to the training process or the inference process, so we have analyzed both situations. We hope it helps clarify any confusion.
> - **1. During training:** Significant differences between the reference pose and the candidates in the pose and alignment pools can actually benefit training by enhancing the model's robustness. Different poses enable the model to understand the difference between complex reference image inputs and driven pose video inputs. For example, in the first row of Figure 1 (original submission *i.e.*, the teaser), we use a human skeleton to drive a limb-less character. To achieve such capability, we need to simulate extreme scenarios during training. Therefore, when the reference pose differs significantly from the candidates in the pose pool and alignment pool during training, it enhances the robustness of the model.
> - **2. During inference:** Even when the reference pose differs significantly from the candidates in the pose and alignment pools, our model is still able to produce reasonable results, which is one of the core challenges addressed in this paper. Our pose pool and alignment pool are designed to encompass a wide range of local deformations, while the IPI module focuses on implicit motion modeling. This combination allows the model to learn generalized motion patterns from videos, rather than being constrained to specific actions. Thus, regardless of the input driver video or its corresponding pose, Animate-X ensures stable and reliable generation without excessive collapse.
>
> ***
>
> **Comment 4: Could aligning the driving pose to a "standard" one in the pose pool further improve generation quality?**
>
> Yes. Aligning the driving pose to a "*standard*" one can further improve generation quality. This is because the "*aligning*" operation simplifies the complexity of the animation process, making it easier for the model to generate accurate results.
>
> ***
>
> **Comment 5: Consider adding a difficulty level split for A$^2$Bench.**
>
> Thanks for your valuable suggestion. We have added the difficulty level split for Animate-X. As shown in Figure 10 (response letter), we categorized the videos in A$^2$Bench into three difficulty levels: Level 1, Level 2, and Level 3. The classification is based on their appearance characteristics.
> - **First**, we classify characters that have body shapes and other appearance features similar to humans, as shown in the first row of Figure 10 (response letter), into the easiest, Level 1 category. These characters are generally simpler to drive, produce fewer artifacts, and have better motion consistency.
> - **In contrast**, characters that maintain more distinct structural features from humans, such as dragons and ducks in the third row of Figure 10 (response letter), are classified into the most difficult Level 3 category. These characters often preserve their original structures (*e.g.*, a duck's webbed feet and wings), which makes balancing identity preservation and motion consistency more challenging. To ensure identity preservation, the consistency of motion may be compromised, and vice versa. Additionally, images involving interactions between characters, objects, environments, and backgrounds are also placed in Level 3, as they increase the difficulty for the model to distinguish the parts that need to be driven from those that do not.
> - **Videos in between these two categories**, like those in the second row of Figure 10 (response letter), are classified as Level 2. These characters often strike a good balance between anthropomorphism and their original form, making them easier to animate with better motion consistency than Level 3 characters and more interesting results than Level 1 characters.
>
> ***
>
> **Comment 6: Most styles in the A$^2$Bench benchmark are "3D render style"; the benchmark should include a wider variety of visual styles. In the supplementary materials, the authors show results in various styles, yet most styles in A$^2$Bench are in "3D render style." Would it be possible to add a "style trigger word" in the prompt template to diversify and strengthen the benchmark?**
>
> Following your suggestions, we have added style trigger words such as *"Watercolor Painting"*, *"Cyberpunk Style"*, *"Van Gogh"*, *"Ukiyo-E"*, *"Pixel Art"*, and so on. Some results are shown in Figure 11 (response letter), which indeed enriches the benchmark and strengthens its diversity. Please see `(.zip/for_reviewer_feUz/more_style/xxx.mp4)` for video results. Thank you for your valuable suggestions.

---

> > ### Comment · Reviewer_feUz · 2024-11-28
> > **Further question about comment 3.**
> >
> > My question in comment 3 is related to the question in comment 1. In comment 1, I wanted to understand the pose statistics in the augmentation pool, and in this comment, I want to know whether Animate-X can only handle poses from a specific domain (which might be more diverse than previous methods).
> >
> > For example, if all poses in the augmentation pool have legs ranging from a to b pixels and arms from c to d pixels, with height/width ratios between e and f, what would happen if the user provides a driving video where the character’s legs are much longer than b, the arms are shorter than c, and the height/width ratio is far outside the range e to f?
> >
> > Since most characters in the $A^2$ Bench have a similar height/width ratio, the author(s) should provide the static values for the augmentation pose pool and include some visualization results demonstrating the model’s robustness on driving videos where the poses do not lie within the augmentation pose pool.

---

> > ### Comment · Reviewer_feUz · 2024-11-28
> > **Further question about comment 5.**
> >
> > I appreciate the effect of the difficulty level split provided in the response letter. Could the author(s) please provide additional evaluation results (maybe 1~2 methods) for each subset?

---

> ### Author Response · Authors · 2024-11-25
>
> Dear Reviewer feUz,
>
> Thank you again for the great efforts and valuable comments. We hope you find the response satisfactory. As the discussion phase is about to close, we are eagerly looking forward to hearing from you regarding any further feedback. We will be more than happy to address any additional concerns you may have.
>
> Best,
>
> Animate-X Authors

---

> ### Author Response · Authors · 2024-11-28
> **Response for the further question about selection of $I^p_{anchor}$**
>
> Thank you for your response and further discussion.
>
> The anchor poses are chosen from the entire training set. To address your next question about “*If they are randomly selected from the entire training set, how does the static distribution of the rescaling ratio (e.g., the shoulder length of $I^p_{anchor}$ divided by the shoulder length of $I^p$) look like?*” we conducted the following statistical analysis.
>
> First, we randomly sampled a driven pose $I^p$ and then traversed the entire pose pool, treating each pose in the pool as an anchor pose to calculate the rescaling ratio. We repeated this process 10 times. Finally, we divided the range from 0.001 to 10 into 10 intervals, counting  the proportion of rescaling ratios that fell within each interval. In addition to the shoulder length that you mentioned, we also analyzed the proportions of other important parts like body length, upper arm length, lower arm length, upper leg length, lower leg length.
>
> | Interval     | Shoulder Lenght | Body Length | Upper Arm Length | Lower Arm length | Upper Leg Length | Lower Leg length |
> |--------------|-----------------|-------------|------------------|------------------|------------------|------------------|
> | [0.001, 0.1) | 0.19%           | 0.14%       | 0.05%            | 0.08%            | 0.05             | 0.81%            |
> | [0.1, 0.3)   | 1.52%           | 5.73%       | 4.04%            | 3.22%            | 0.59%            | 4.60%            |
> | [0.3, 0.5)   | 12.21%          | 18.57%      | 15.28%           | 7.63%            | 4.26%            | 5.65%            |
> | [0.5, 0.7)   | 15.33%          | 16.93%      | 12.97%           | 7.54%            | 12.02%           | 9.61%            |
> | [0.7, 1.0)     | 20.07%          | 18.48%      | 17.15%           | 11.35%           | 24.86%           | 19.53%           |
> | [1.0, 1.5)   | 22.09%   | 18.63%    | 17.56%      | 15.38%     | 27.90%       | 24.89%      |
> | [1.5, 2)     | 10.07%    | 8.34%     | 7.93%      | 11.73%      | 14.31%       | 14.47%    |
> | [2.0, 3.0)       | 9.75%           | 6.52%       | 7.73%            | 16.19%           | 11.83%           | 15.28%           |
> | [3.0, 6.0)       | 6.33%           | 6.28%       | 10.93%           | 18.40%           | 2.73%            | 4.30%            |
> | [6.0, 10.0)      | 2.43%           | 0.37%       | 6.37%            | 8.47%            | 1.45%            | 0.85%            |
>
> It can be observed that the overall distribution covers a wide range (from 0.001 to 10.0), which allows the model to learn poses of various characters, encompassing non-human subjects. We will include the above statistical information in the supplementary materials. Thank you for the valuable comments.
>
>
> If you have any further questions or concerns, please do not hesitate to reach out to us for further discussion.

---

> ### Author Response · Authors · 2024-11-28
> **Response for *further question about comment 5***
>
> Following your suggestion, we evaluate the results of Animate-X and UniAnimate for each subset and present the results below:
>
> |                   | PSNR* ↑ | SSIM ↑ | L1     ↓  | LPIPS ↓ | FID  ↓ | FID-VID ↓ | FVD   ↓  |
> |-------------------|-------|-------|----------|--------|-------|---------|---------|
> | Animate-X-level1  | 13.96 | 0.461 | 9.67E-05 | 0.418  | 24.24 | 31.37   | 681.53  |
> | Animate-X-level2  | 13.74 | 0.457 | 9.82E-05 | 0.429  | 26.12 | 32.19   | 693.63  |
> | Animate-X-level3  | 13.17 | 0.442 | 1.11E-04 | 0.437  | 27.34 | 35.64   | 721.41  |
> | UniAnimate-level1 | 11.93 | 0.413 | 1.14E-04 | 0.521  | 42.39 | 52.14   | 1120.45 |
> | UniAnimate-level2 | 11.89 | 0.408 | 1.20E-04 | 0.526  | 46.27 | 58.53   | 1147.34 |
> | UniAnimate-level3 | 10.91 | 0.379 | 1.35E-04 | 0.549  | 56.58 | 65.39   | 1204.53 |
>
> We can see that as the difficulty increases, each evaluation result shows a decline. Thank you for your feedback and we will include the above results in the supplementary materials.

---

> ### Author Response · Authors · 2024-11-28
> **Response for *Further question about comment 3.***
>
> Thanks. Since we are unable to update the PDF to include the visual results at this stage, we describe them as clearly as possible in text. If there are any misunderstandings, please point them out.
>
> **First**, we have included the distribution statistics of the static values for the augmentation pose pool (as mentioned in the *Response for the further question about selection of $I^p_{anchor}$*). **Then**, to investigate the model's performance on driven videos with sufficiently large differences from these augmentation strategies, we collect such cases as input: The driven video features the pose images with a very high body but an extremely slim physique ( the height/width ratio > f). The arms are quite short (< c), while the legs are long (>b). Our method still successfully animate the reference image with the motion of the driven video.
>
> The model maintains this robustness because it learns the local motion patterns. As the reviewer mentioned, although cases like "*leg(longer than b) + arm(shorter than c) + ratio(far outside the range e to f)*" are not seen in the training data, the model learns to recognize the pixel changes when a part, like the arm, is significantly shortened. The same applies to the legs and the ratio. When these components are combined, the diffusion model is able to handle them properly.  Additionally, for extreme cases, such as very long arms, there is always a pixel boundary (*i.e.*, the pixel boundary of each frame or image). However, our EPI covers most of this range, and any missing parts can be addressed by the model’s inherent generative capability and the IPI.
>
> If possible, we will add the visual results and analysis to the final supplementary materials, which is limited by current rules. Thanks for your suggestions.

---

> ### Comment · Reviewer_feUz · 2024-11-29
>
> Thank you to the author(s) for the detailed explanation. Since the replies address most of my questions, I have decided to increase my score.

---

> ### Author Response · Authors · 2024-11-29
> **Thanks a lot for the suggestions and valuable comments!**
>
> Thanks a lot for the suggestions and valuable comments! We are pleased to know that our responses have addressed your questions. We appreciate for your decision to raise the score!

---

### Official Review · Reviewer_mbHE · 2024-11-03

**Soundness:** 2
**Presentation:** 2
**Contribution:** 2
**Rating:** 6
**Confidence:** 5

**Summary:**

This paper focuses on the animation of non-human characters, addressing two main issues:
1)Sole pose skeletons lack image-level details.
2)Pose alignment in the self-driven reconstruction training strategy.
To resolve these issues, the paper introduces a Pose Indicator, comprising an Implicit Pose Indicator and an Explicit Pose Indicator. Experimental results demonstrate that the proposed Animate-X achieves effective performance in character animation.

**Strengths:**

1.The authors introduce  A2Bench, which is helpful for the evaluation of character animation.
2.Both qualitative and quantitative experiments are conducted to evaluate the performance of the proposed method.

**Weaknesses:**

1.Some parts of the writing can be quite confusing, words and sentences are bad orgnized. For example, in P5 L260, what exactly is in the pose pool? And how is it aligned with the reference?
2.The dataset includes 9,000 independently collected videos. Could you analyze these videos, and did other baselines use the same data for training? If not, could this lead to an unfair comparison?
3.The authors first identify the weaknesses of previous methods as a conflict between identity preservation and pose control. They further expand on this point by highlighting two specific limitations: the lack of image-level details in sole pose skeletons and pose alignment within the self-driven reconstruction training strategy. However, while the authors clearly state that differences in appearance between characters and humans can negatively impact animation, learning image-level details seems to contradict their viewpoint  "sole pose skeletons lack image-level details", making this contribution appear more like a forced addition.
4.Additionally, the visualization in Figure 7 provided by the authors also supports w3. The inclusion or exclusion of the IPI appears to have minimal impact on the motion of the Ref image, and with IPI, part of the foot in the Ref image is even missing. This raises doubts about the effectiveness of the IPI module and seems inconsistent with the authors' stated motivation.
5.Pose augmentation has already been widely explored in existing methods, such as MimicMotion, which makes the innovation in this paper insufficient.
6.This paper lacks comparisons with similar methods, such as MimicMotion, which makes the experimental results less convincing.
[1]MimicMotion: High-Quality Human Motion Video Generation with Confidence-aware Pose Guidance

**Questions:**

See Weakness. If the authors can address all my concerns, I am willing to raise the score.

---

> ### Author Response · Authors · 2024-11-20
> **Thank you for your valuable feedbacks.**
>
> We sincerely thank **Reviewer #2 mbHE** for acknowledging the *"introduced A²Bench, the qualitative and quantitative experiments presented in our work"*. We have re-uploaded our supplementary materials, which include the complete responses (at .zip/Animate-X_rebuttal_response_letter.pdf) along with the relevant figures and tables. The response letter is also contained in the main paper, after page 25. Below, we have addressed each question in detail and hope to clarify any concerns.
>
> **Comment 1: Some parts of the writing can be quite confusing, words and sentences are bad organized. For example, in P5 L260, what exactly is in the pose pool  (Answer 1.1)? And how is it aligned with the reference?  (Answer 1.2)**
>
> **Answer 1.1.** The pose pool mentioned in P5 L260 consists of all the unenhanced pose images extracted from our training dataset. Specifically, we use DWPose as the pose extractor to obtain skeleton images with a black background from the training videos.
>
> **Answer 1.2.** We have provided a detailed explanation of the pose pool and alignment process in Appendix A and Figure 4 (response letter). The alignment process can be organized into the following steps:
> - **Step1:** Given a driving pose $I^p$, we randomly select an anchor pose $I^p_{anchor}$ from the pose pool.
> - **Step2:** We then calculate the proportion of each body part between these two poses. For example, the shoulder length of $I^p_{anchor}$ divided by the shoulder length of $I^p$ might be 0.45, and the leg length of $I^p_{anchor}$ divided by the leg length of $I^p$ might be 0.53, and so on.
> - **Step3:** We multiply each body part of the driven pose (*i.e.*, $I^p$) by the corresponding ratio (*e.g.*, 0.45, 0.53, *etc.*) to obtain the aligned pose (*i.e.*, $I^p_n$).
>
> ***
>
> **Comment 2: The dataset includes 9,000 independently collected videos. Could you analyze these videos (Answer 2.1), and did other baselines use the same data for training  (Answer 2.2)? If not, could this lead to an unfair comparison (Answer 2.3)?**
>
> Thanks for your valuable comments. First, we would like to clarify that we have demonstrated the improvements in our approach stem from the IPI and EPI modules through the extensive and fair ablation experiments. Next, we will address each question in detail.
>
> **Answer 2.1.** Following the commonly used public human animation TikTok datasets which consists of videos downloaded from TikTok, we additionally collect 9,000 TikTok-like videos. The distribution of the additional data is similar to the TikTok dataset, primarily consisting of human dance videos.
>
> **Answer 2.2.** We notice that other baselines have also used their own collected data for model training. For example, UniAnimate uses 10,000 internal videos. Despite using more data than we did, Animate-X still improves the performance substantially, suggesting that these gains stem from the design of our modules rather than the data.
>
> **Answer 2.3.** Data is also the essential contribution of each respective work. The use of independently collected videos, including in our work, is transparently explained in the papers and has become a well-established convention in prior researches. **To address potential concerns**, we have trained our Animate-X solely on the public TikTok and Fashion benchmarks, **without incorporating any extra videos**. We have conducted the same experiments as presented in Table 1 (original submission), and reported results marked by # in **Table IV**. As shown in **Table IV**, our method still outperforms other approaches, which further demonstrates that the improvements in Animate-X are driven by the IPI and EPI modules, rather than the use of additional training data.

---

> ### Author Response · Authors · 2024-11-20
> **Response (2)**
>
> | **Method**                | **PSNR*** ↑   | **SSIM** ↑   | **L1** ↓         | **LPIPS** ↓   | **FID** ↓       | **FID-VID** ↓  | **FVD** ↓       |
> |---------------------------|---------------|--------------|------------------|---------------|-----------------|----------------|-----------------|
> | Moore-AnimateAnyone       | 9.86          | 0.299        | 1.58E-04         | 0.626         | 50.97           | 75.11          | 1367.84         |
> | MimicMotion (*ArXiv24*)   | 10.18         | 0.318        | 1.51E-04         | 0.622         | 122.92          | 129.40         | 2250.13         |
> | ControlNeXt (*ArXiv24*)   | 10.88         | 0.379        | 1.38E-04         | 0.572         | 68.15           | 81.05          | 1652.09         |
> | MusePose (*ArXiv24*)      | 11.05         | 0.397        | 1.27E-04         | 0.549         | 100.91          | 114.15         | 1760.46         |
> | Unianimate (*ArXiv24*)    | *11.82*       | *0.398*      | *1.24E-04*       | *0.532*       | *48.47*         | *61.03*        | *1156.36*       |
> | **Animate-X #**      | 13.46         | 0.441        | 1.19E-04         | 0.468         | 37.76           | 40.19          | 933.43          |
> | **Animate-X**             | **13.60**     | **0.452**    | **1.02E-04**     | **0.430**     | **26.11**       | **32.23**      | **703.87**      |
>
> **Table IV:** Quantitative comparisons with SOTAs on A²Bench. The best and second-best results for each column are **bold** and *italicized*, respectively.
>
> ***
>
> **Comment 3: The authors first identify the weaknesses of previous methods as a conflict between identity preservation and pose control. They further expand on this point by highlighting two specific limitations: the lack of image-level details in sole pose skeletons and pose alignment within the self-driven reconstruction training strategy. However, while the authors clearly state that differences in appearance between characters and humans can negatively impact animation, learning image-level details seems to contradict their viewpoint "sole pose skeletons lack image-level details",  making this contribution appear more like a forced addition.**
>
> We disagree with this comment. "*sole pose skeletons lack image-level details* and "*learning image-level details*" are not contradictory but rather represent a cause-and-effect relationship. As shown in Figure 7 (response letter), previous methods extract only pose skeletons from original driving videos. The process can be represented as
>
> > video → pose skeletons → results.
>
> These pose skeletons lack image-level motion-related details, *i.e.*, motion-induced deformations (*e.g.*, body part overlap and occlusion). These details play a crucial role in enhancing character animation, since personification cartoon characters have more unpredictable movement patterns compared to humans. Therefore, we design the IPI module specifically to extract these image-level motion-related details. The process can be represented as:
> > **Step 1:** (*as same as the previous method*) video → pose images
> > **Step 2:** video → IPI → image-level motion-related features
> > **Step 3:** pose images + image-level motion-related features → results
>
> **Moreover**, the introduction of our IPI module is a core contribution of this paper which is not "*a forced addition*". In previous approaches, temporal information in driven videos was derived solely from multi-frame pose skeletons, often set against pure black backgrounds. The original RGB videos were discarded during the training process. While this method works well for human animation, where carefully designed pose skeletons align perfectly with human joints, it falls short for anthropomorphic characters whose skeletons differ significantly from humans. Thus, pose skeletons alone can **NOT** provide sufficient driving guidance, as they lack the motion-related details found only in the original driving video. This is where our IPI module makes a difference, extracting these richer details from the original video to improve the generalization of motion representation modeling.

---

> ### Author Response · Authors · 2024-11-20
> **Response (3)**
>
> **Comment 4: Additionally, the visualization in Figure 7 provided by the authors also supports w3. The inclusion or exclusion of the IPI appears to have minimal impact on the motion of the Ref image, and with IPI, part of the foot in the Ref image is even missing. This raises doubts about the effectiveness of the IPI module and seems inconsistent with the authors' stated motivation.**
>
> Thanks for the comment. The *"missing foot"* is caused by the video not being fully displayed in our submission, rather than an issue with our IPI module. We have added more frames of the video in Figure 8 (response letter). Please refer to the video result in `(.zip/for_reviewer_mbHE/full_frame_of_figure7.mp4)`. As shown in Figure 8 (response letter), in the initial frames, the foot is present and highly consistent with the reference image. Subsequently, the driven pose image begins to perform a leg-merging motion, with the distance between the legs gradually decreasing. To allow the anthropomorphic bamboo character to follow this motion, it also gradually merges its legs, giving the appearance of the *"missing foot"*.
>
> ***
>
> **Comment 5: Pose augmentation has already been widely explored in existing methods, such as MimicMotion, which makes the innovation in this paper insufficient.**
>
> The primary contribution of our work is animating anthropomorphic figures using two new modules, IPI and EPI, which go beyond simple *"pose augmentation"*. Pose augmentation is a training strategy and is not exclusive to any specific method. By itself, it cannot solve the animation issue in our work. The IPI and EPI modules designed to handle figures beyond human and human pose are novel to address the specific challenges in animating anthropomorphic figures. We then provide a detailed explanation of the concept beyond "*Pose Augmentation*". Please refer to Figure 4 (response letter) or Figure 8 (original submission) for an illustration of the following process:
> - **Step1:** We first construct the pose pool using the DWPose extractor. The pose pool consists of pose skeletons (*i.e.*, pose images).
> - **Step2:** Given a driving pose $I^p$, we randomly select an anchor pose $I^p_{anchor}$ from the pose pool.
> - **Step3:** We then calculate the proportion of each body part between these two poses. For example, the shoulder length of $I^p_{anchor}$ divided by the shoulder length of $I^p$ might be 0.45, and the leg length of $I^p_{anchor}$ divided by the leg length of $I^p$ might be 0.53, and so on.
> - **Step4:** We multiply each body part of the driven pose (*i.e.*, $I^p$) by the corresponding ratio (*e.g.*, 0.45, 0.53, *etc.*) to obtain the aligned pose (*i.e.*, $I^p_n$).
> - **Step5:** Then we define a set of keypoint rescaling operations, including modifying the length of the body, legs, arms, neck, and shoulders, altering face size, adding or removing specific body parts, *etc*. These transformations are stored in a rescale pool.
> - **Step6:** We apply the selected transformations on the aligned pose $I^p_{realign}$ to obtain the final transformed poses $I^p_n$.
>
> ***
>
> **Comment 6: This paper lacks comparisons with similar methods, such as MimicMotion, which makes the experimental results less convincing. [1]MimicMotion: High-Quality Human Motion Video Generation with Confidence-aware Pose Guidance.**
>
> We have already conducted:
> **(1) Quantitative comparisons** with MimicMotion in Tables 1, 2, 7, and 8 in the original submission.
> **(2) Qualitative comparisons** with MimicMotion in Figure 5 and the videos in the original *Supplementary Materials*.
> **(3) The user study comparison** with MimicMotion in Table 3 in the original submission.
> For your convenience, we highlight and summary these results below.
>
> | **Method**                | **PSNR*** ↑   | **SSIM** ↑   | **L1** ↓         | **LPIPS** ↓   | **FID** ↓       | **FID-VID** ↓  | **FVD** ↓       |
> |---------------------------|---------------|--------------|------------------|---------------|-----------------|----------------|-----------------|
> | MimicMotion (*ArXiv24*)   | 10.18         | 0.318        | 1.51E-04         | 0.622         | 122.92          | 129.40         | 2250.13         |
> | **Animate-X**             | **13.60**     | **0.452**    | **1.02E-04**     | **0.430**     | **26.11**       | **32.23**      | **703.87**      |
>
> **Table V:** Quantitative comparisons with MimicMotion on A²Bench with the rescaled pose setting. The best results for each column are **bold**.

---

> ### Author Response · Authors · 2024-11-20
> **Response (4)**
>
> | **Method**                | **PSNR*** ↑   | **SSIM** ↑   | **L1** ↓         | **LPIPS** ↓   | **FID** ↓       | **FID-VID** ↓  | **FVD** ↓       |
> |---------------------------|---------------|--------------|------------------|---------------|-----------------|----------------|-----------------|
> | MimicMotion (*ArXiv24*)   | 12.66         | 0.407        | 1.07E-04         | 0.497         | 96.46           | 61.77          | 1368.83         |
> | **Animate-X**             | **14.10**     | **0.463**    | **8.92E-05**     | **0.425**     | **31.58**       | **33.15**      | **849.19**      |
>
> **Table VI:** Quantitative comparisons with MimicMotion on A²Bench in the self-driven setting. The best results for each column are **bold**.
>
> ***
>
> | **Method**                | **Moore-AA** | **MimicMotion** | **ControlNeXt** | **MusePose** | **Unianimate** | **Animate-X**      |
> |---------------------------|--------------|-----------------|-----------------|--------------|----------------|--------------------|
> | **Identity preservation ↑** | 60.4%        | 14.8%           | 52.0%          | 31.3%        | 43.0%          | **98.5%**          |
> | **Temporal consistency ↑**  | 19.8%        | 24.9%           | 36.9%          | 43.9%        | 81.1%          | **93.4%**          |
> | **Visual quality ↑**        | 27.0%        | 17.2%           | 40.4%          | 40.3%        | 79.3%          | **95.8%**          |
>
> **Table VII:** User study results. The best results for each metric are **bold**.
>
> ***
>
> | **Method**                | **L1** ↓      | **PSNR*** ↑   | **SSIM** ↑   | **LPIPS** ↓   | **FVD** ↓       |
> |---------------------------|---------------|---------------|--------------|---------------|-----------------|
> | MimicMotion (*ArXiv24*)   | 5.85E-04      | 14.44         | 0.601        | 0.414         | 232.95          |
> | **Animate-X**             | **2.70E-04**  | **20.77**     | **0.806**    | **0.232**     | **139.01**      |
>
> **Table VIII:** Quantitative comparisons with MimicMotion on the TikTok dataset. The best results for each metric are **bold**.
>
> ***
>
> | **Method**                | **PSNR*** ↑   | **SSIM** ↑   | **LPIPS** ↓   | **FVD** ↓       |
> |---------------------------|---------------|--------------|---------------|-----------------|
> | MimicMotion (*ArXiv24*)   | 27.06         | 0.928        | 0.036         | 118.48          |
> | **Animate-X**             | **27.78**     | **0.940**    | **0.030**     | **79.4**        |
>
> **Table IX:** Quantitative comparisons with MimicMotion on the Fashion dataset. The best results for each metric are **bold**.

---

> ### Author Response · Authors · 2024-11-25
>
> Dear Reviewer mbHE,
>
> Thank you again for the great efforts and valuable comments. We hope you find the response satisfactory. As the discussion phase is about to close, we are eagerly looking forward to hearing from you regarding any further feedback. We will be more than happy to address any additional concerns you may have.
>
> Best,
>
> Animate-X Authors

---

> > ### Comment · Reviewer_mbHE · 2024-11-29
> >
> > Thank you for your response, the reviewer acknowledges that the responses to W1 and W2 are satisfactory and address the concerns adequately.
> >
> > However, the responses for W3 and W5 remain insufficient to fully address my concerns:
> >
> > Regarding W3:
> >
> > The use of RGB patches has already been proposed in existing portrait animation methods, such as X-Portrait and Meg-Actor. However, these methods are only capable of learning appearance-related motion within the human in-domain, leading to significant content leakage issues and a lack of adaptability to out-domain scenarios.
> > Therefore, I believe it is unreasonable to introduce appearance feature learning in this context, as the appearance features of each character are distinct, and the corresponding appearance-relative motion features also differ.
> >
> > Regrading W5:
> >
> > The effectiveness of pose image augmentation has already been demonstrated in MimicMotion, which diminishes the insight provided by this paper. Perhaps an ablation study comparing the data augmentation methods in this paper and MimicMotion could be provided?

---

> > > ### Author Response · Authors · 2024-11-29
> > >
> > > Thank you for your response and valuable comments. We are pleased to know that the responses to W1 and W2 are satisfactory and adequately address your concerns. Regarding your concerns about W3 and W5, we would like to provide the following explanations:
> > >
> > > **For W3:**
> > >
> > > We have carefully studied X-Portrait and Meg-Actor, and we found that they employ Control Modules (in X-Portrait) and DrivenEncoder (in Meg-Actor) to directly extract features from RGB patches without adding any additional constraints. As you mentioned, this approach can easily lead to the inclusion of appearance features. In contrast, our approach first utilizes a pre-trained CLIP encoder to extract CLIP features from RGB patches, which prevents the extraction of appearance features **at the initial step**. Furthermore, we use DWpose data containing only motion information as a guiding signal (i.e., Q) to filter the CLIP features, allowing us to isolate motion-relative features and mitigate the influence of appearance features **in a second step**.
> > >
> > > Additionally, regarding your concern about generalization (i.e., "*as the appearance features of each character are distinct, the corresponding appearance-relative motion features also differ*"), the used CLIP image encoder is trained on a large-scale dataset, which equips it to handle characters with different appearances. With the support of our proposed IPI, the features extracted from images by our model are solely related to motion, even if *the appearance of each character is distinct*, which greatly benefits our training process.
> > >
> > > If you have any further questions or specific cases you would like to test, we would be more than happy to address them and conduct the necessary tests.
> > >
> > > ***
> > >
> > > **For W5:**
> > >
> > > We have read through the entire MimicMotion paper and checked its open source code many times, but we did not find the "*pose image augmentation*" in MimicMotion and only find the "pose uncertainty estimation", which is different from our proposed EPI. The pose uncertainty estimation is strictly aligned with human poses, which enables the model to generate clear representations of human hands and highlight other key parts of the human body. Our EPI explicitly implements pose augmentation to enhance the model's adaptability to non-human subjects, such as cartoon characters with long arms or those without limbs. If we have any omissions about the "*pose image augmentation*" in MimicMotion that we hope the reviewer can correct, we will conduct corresponding comparative experiments.
> > >
> > > Anyway, to the best of our ability, we provide the ablation study that we could think of in response to the reviewers' request. Specifically, we replaced our EPI with the "**p**ose **u**ncertainty **e**stimation (PUE)" from MimicMotion. The corresponding results are presented in the table below.
> > >
> > > |          | PSNR* ↑ | SSIM ↑ | LPIPS ↓ | FID ↓  | FVD ↓   |
> > > |----------|---------|--------|---------|--------|---------|
> > > | with PUE | 11.95   | 0.404  | 0.526   | 53.83  | 1031.84 |
> > > | with EPI (ours) | 13.30   |  0.433   |  0.454  | 32.56 | 893.31  |
> > >
> > > From the results, we can see that PUE from MimicMotion provides the limited improvement to our new task. We appreciate the efforts of MimicMotion in improving pose accuracy. This is a highly effective approach that has greatly inspired us, and we will highlight this point in our revised version. Thank you for your contribution to improving the quality of our manuscripts.

---

> > > > ### Comment · Reviewer_mbHE · 2024-12-02
> > > >
> > > > Thank you for your response.
> > > >
> > > > For W3
> > > >
> > > > CLIP features exhibit highly aggressive appearance-related semantic information, and many methods leverage them as injected appearance features.
> > > >
> > > > Authors highlight the need to learn motion features tied to appearance while simultaneously learning motion features independent of appearance. This is contradictory and lacks persuasiveness.
> > > >
> > > > For W5:
> > > >
> > > > In MimicMotion/mimicmotion/dwpose/preprocess.py line 43.

---

> > > > > ### Author Response · Authors · 2024-12-02
> > > > >
> > > > > Thanks for your response.
> > > > >
> > > > > **For W3:**
> > > > >
> > > > > 1. Our discussion on CLIP’s ability to avoid appearance-based information extraction refers specifically to X-Portrait and Meg-Actor. They use raw RGB data, which fully preserves all appearance information.
> > > > >
> > > > > 2. The CLIP encoder does retrieve richer information, including the appearance information, from driven videos. But in IPI, we use DWPose data containing only motion information as a guiding signal (i.e., Q) to filter the CLIP features, allowing the model to isolate motion-relative features and mitigate the influence of appearance features. This is a trade-off based on the model’s learning capability, where the newly learned motion representation has stronger modeling ability for general motion. In contrast, pure motion features represented by pose skeletons are completely decoupled from video appearance. However, their strong correspondence with the human body makes it challenging to generalize to non-human subjects. This is one of the reasons we aim to enhance motion features. Our ablation experiments show that using the IPI and an appropriate training strategy does not cause the model to incorporate appearance information from the reference video.
> > > > >
> > > > > ***
> > > > >
> > > > > **For W5:**
> > > > >
> > > > > First, the purpose of this widely used function (MimicMotion/mimicmotion/dwpose/preprocess.py line 43) is to **align** the driven pose with the reference image, while our EPI serves as an augmentation during the training process to **prevent alignment**. Second, this codebase only includes inference codes, and the MimicMotion paper (Page 4, Sec. 3.2, arXiv:2406.19680) does not mention the use of this function during training or inference. Since the training data is already aligned (*i.e.*, the reference image is randomly sampled from the same video), we believe the relevance of this function to training is minimal. Therefore, we consider these as two distinct contributions from different perspectives, which are not in conflict.

---

> > > > > > ### Comment · Reviewer_mbHE · 2024-12-02
> > > > > >
> > > > > > Thank you for your detailed reply, i am willing to raise my score.
> > > > > >
> > > > > > However, while the authors emphasize the performance gain, the reviewer is concerned about the novelty as echoing existing tricks in new tasks is not a good way to advance the research community.

---

> > > > > > > ### Author Response · Authors · 2024-12-02
> > > > > > >
> > > > > > > Thanks a lot for the suggestions and valuable comments! We appreciate for your decision to raise the score! We are confident in our new task and approach we have proposed. We will release more details about this work at an appropriate time to further advance community development.

---

### Official Review · Reviewer_aHUH · 2024-11-03

**Soundness:** 2
**Presentation:** 2
**Contribution:** 2
**Rating:** 8
**Confidence:** 5

**Summary:**

This work presents an animation framework capable of animating anthropomorphic characters, along with an accompanying benchmark for animated anthropomorphic characters. Specifically, the framework introduces an Implicit Pose Indicator and an Explicit Pose Indicator to provide rich pose guidance.

**Strengths:**

1. The visual results of Animate-X demonstrate notable improvements across various characters compared to existing animation methods.
2. Comprehensive experiments and ablation studies are presented.

**Weaknesses:**

1. No video samples from A2Bench are provided; only selected frames are shown in the paper. Given that the generated videos still struggle with maintaining strict logic and good spatial and temporal consistency, I question the rationale for using T2I + I2V to generate benchmark videos. Additionally, the benchmark lacks detailed information, such as video length and frame rate. Were any additional motion prompts used to generate videos from images? If so, what is their diversity and complexity?
2. The necessity of a pose pool and the selection of an anchor pose image need clarification. What operations are involved in the "align" process, specifically regarding translation and rescaling? Why not use random translation and rescaling instead of relying on an anchor pose image?
3. The effectiveness of the Implicit Pose Indicator (IPI) is also in question. The motivation for the IPI is that sparse keypoints lack image-level details, while IPI aims to retrieve richer information. However, Tables 7 and 8 indicate that Animate-X achieves comparable performance to Animate-Anyone and UniAnimate on human videos. This suggests that the IPI does not provide any benefits for human animation.

**Questions:**

Please address the concerns in the weakness.

---

> ### Author Response · Authors · 2024-11-20
> **Thank you for your valuable feedbacks.**
>
> We sincerely thank **Reviewer #1 aHUH** for acknowledging the *``notable improvements of Animate-X''* and the *``comprehensive experiments and ablation studies presented in our work''*. We have re-uploaded our supplementary materials, which include the complete responses (at .zip/Animate-X_rebuttal_response_letter.pdf) along with the relevant figures and tables. The response letter is also contained in the main paper, after page 25. Below, we have addressed each questions in detail and hope to clarify any concerns.
>
> ***
>
> **Comment 1: No video samples from A$^2$Bench are provided; only selected frames are shown in the paper. Given that the generated videos still struggle with maintaining strict logic and good spatial and temporal consistency, I question the rationale for using T2I + I2V to generate benchmark videos.**
>
>  Thanks. We have provided video samples of A$^2$Bench in the updated *Supplementary Materials* (.zip/for\_reviewer\_aHUH/xxx.mp4). We kindly invite the reviewer to check these videos. Below, we address the reviewer's concerns regarding *``strict logic''* and *``good spatial and temporal consistency''* using T2I + I2V:
> - **1. Strict logic:** The choice to use T2I models stems from a clear need: current T2V models often struggle with imaginative and logically complex inputs, such as "*personified refrigerators*" or "*human-like bees*". T2I models offer strict logic and imagination in these scenarios, allowing to generate reasonable cartoon characters as the ground-truth. To prove this point, as shown in Table I, we assessed the semantic accuracy of A$^2$Bench using CLIP scores, which are commonly used to evaluate whether the semantic logic of images and text is strictly aligned (*i.e.*, Does the generated ``*human-like bee*'' maintain the visual essence of a bee while seamlessly incorporating human-like features, such as hands and feet?). For comparison, we also evaluate the publicly available TikTok and Fashion datasets using the same metric. These experimental results demonstrate that A$^2$Bench achieves the highest level of strict logical alignment. **Furthermore**, we input the images from A$^2$Bench into a multimodal large language model (MLLM) with logical reasoning, such as QWen, to conduct a logical analysis of the visual outputs generated by the T2I model. The results, shown in Figure 1 (response letter), reveal that the image descriptions answered by the MLLM closely aligns with our input prompts, which verifies again that the data in A$^2$Bench maintains strict logic.
> - **2. Good spatial and temporal consistency:** We have incorporated several metrics from VBench, such as *Background Consistency*, *Motion Smoothness*, *Aesthetic Quality*, and *Image Quality*, to evaluate the spatial and temporal consistency of the videos in A²Bench. As shown in **Table I**, A$^2$Bench outperforms the TikTok dataset across all metrics and achieves comparable scores to the Fashion dataset, both of which are collected from real-world scenarios. This demonstrates that the videos generated by our method exhibit a similar level of spatial and temporal consistency to real-world videos.
>
>  In summary, to our best knowledge, T2I+I2V is the reasonable and effective solution currently available for automating the production of videos with anthropomorphic cartoon characters. Specifically, the T2I model can understand the prompt and generate well-aligned high-quality images with strict logic, while the I2V model can preserve the identity of the characters in the image and generate videos with good spatial and temporal consistency. Moreover, the T2I step allows human artists to check and make manual modification to the cartoon characters if necessary before generating the videos.
>
> | **Benchmark**  | **CLIP Score** | **Background Consistency** | **Motion Smoothness** | **Aesthetic Quality** | **Image Quality** |
> |-----------------|----------------|----------------------------|------------------------|------------------------|--------------------|
> | TikTok          | *26.92*        | 94.10%                    | 99.05%                | *55.14%*              | *62.54%*          |
> | Fashion         | 20.18          | **98.25%**                | **99.45%**            | 49.62%                | 49.96%            |
> | A²Bench         | **33.24**      | *96.66%*                  | *99.39%*              | **69.86%**            | **69.32%**        |
>
> **Table I:** Quantitative results of different benchmarks. The best and second-best results for each column are **bold** and *italicized*, respectively.

---

> ### Author Response · Authors · 2024-11-20
> **Response (2)**
>
> **Comment 2: Additionally, the benchmark lacks detailed information, such as video length and frame rate (Answer 2.1). Were any additional motion prompts used to generate videos from images (Answer 2.2)? If so, what is their diversity and complexity (Answer 2.3)?**
>
> **Answer 2.1.** Each video in A²Bench is 5 seconds long, with a frame rate of 30 FPS and a resolution of 832 × 1216.
>
> **Answer 2.2.** When generating videos from images, we supplement the prompt in Figure 10 (original submission) regarding spatial relationships, physical logic, and temporal consistency. Examples include: *"reasonable movement"*, *"varied dance"*, and *"continuous dance"*. These prompts further ensure strict logic and good spatial and temporal consistency.
>
> **Answer 2.3.** To guarantee diversity and complexity, for each prompt, we first generate 4 images using 4 different random seeds. Then, for each image, we generate 4 videos. This process ensures both diversity and complexity in the final results. Moreover, as suggested by **Reviewer #3 feUz**, we add style trigger words such as *"Watercolor Painting"*, *"Cyberpunk Style"*, *"Van Gogh"*, *"Ukiyo-E"*, *"Pixel Art"*, and so on. The results are presented in Figure 3 (response letter), which further enhances the diversity and complexity of A$^2$Bench.
>
> ***
>
> **Comment 3: The necessity of a pose pool and the selection of an anchor pose image need clarification (Answer 3.3). What operations are involved in the ''align'' process (Answer 3.1), specifically regarding translation and rescaling (Answer 3.2)? Why not use random translation and rescaling instead of relying on an anchor pose image (Answer 3.3)?**
>
> **Answer 3.1.** As shown in the left half of Figure 8 (original submission) or Figure 4 (response letter), the operations in the ''align'' process are as follows:
> - **Step1:** Given a driving pose $I^p$, we randomly select an anchor pose $I^p_{anchor}$ from the pose pool (two examples are shown in Figure 8.)
> - **Step2:** We then calculate the proportion of each body part between these two poses. For example, the shoulder length of $I^p_{anchor}$ divided by the shoulder length of $I^p$ might be 0.45, and the leg length of $I^p_{anchor}$ divided by the leg length of $I^p$ might be 0.53, and so on.
> - **Step3:** We multiply each body part of the driven pose (*i.e.*, $I^p$) by the corresponding ratio (*e.g.*, 0.45, 0.53, *etc.*) to obtain the aligned pose (*i.e.*, $I^p_n$).
>
> **Answer 3.2.** As shown in the right half of Figure 4 (response letter):
> - **Step4:** (*"rescaling"*) Then we define a set of keypoint rescaling operations, including modifying the length of the body, legs, arms, neck, and shoulders, altering face size, adding or removing specific body parts, *etc.* These operations are stored in a rescale pool.
> - **Step5:** (*"translation"*) We apply the selected rescaling operations on the aligned pose $I^p_{realign}$ to obtain the final transformed poses $I^p_n$.
>
> **Answer 3.3.** As shown in Figure 5 (response letter), the reason for *"not using random translation and rescaling instead of relying on an anchor pose image"* is that random translation and rescaling disrupt the motion guidance originally conveyed by the driven pose image. This issue makes the animation model miss the accurate driving guidance, which diminishes its ability to generate proper animations. In contrast, using anchor pose images maintain harmonious proportions for each body part and preserve the consistency of all motion details.
>
> To prove this point, we **re-trained** our model using pose images obtained through **random** translation and rescaling. The results, presented in Figure 6 (response letter), indicate that the baseline achieves only a marginal improvement (*i.e.*, the content of the reference image only appears in the initial frames, while illogical human characteristics persist throughout). In contrast, our approach delivers satisfactory performance (*i.e.*, it perfectly preserves the cartoon ID of the reference image while adding dynamic motion).
>
> Finally, as shown in **Table II**, quantitative results of ablation study indicate that the "realign" operation plays a crucial role in improving performance, which justifies both the pose pool and the selection of an anchor pose for EPI alignment.

---

> ### Author Response · Authors · 2024-11-20
> **Response (3)**
>
> | **Method**               | **PSNR*** ↑   | **SSIM** ↑   | **L1** ↓       | **LPIPS** ↓   | **FID** ↓     | **FID-VID** ↓ | **FVD** ↓     |
> |--------------------------|---------------|--------------|----------------|---------------|---------------|---------------|---------------|
> | w/o Add in EPI           | 13.28         | 0.442        | 1.56E-04       | 0.459         | 34.24         | 52.94         | 804.37        |
> | w/o Drop in EPI          | 13.36         | 0.441        | 1.94E-04       | 0.458         | *26.65*       | 44.55         | 764.52        |
> | w/o BS in EPI            | 13.27         | 0.443        | 1.08E-04       | 0.461         | 29.60         | 56.56         | 850.17        |
> | w/o NF in EPI            | *13.41*       | *0.446*      | 1.82E-04       | 0.455         | 29.21         | 56.48         | 878.11        |
> | w/o AL in EPI            | 13.04         | 0.429        | *1.04E-04*     | 0.474         | 27.17         | *33.97*       | 765.69        |
> | w/o Rescalings in EPI    | 13.23         | 0.438        | 1.21E-04       | 0.464         | 27.64         | 35.95         | *721.11*      |
> | w/o Realign in EPI       | 12.27         | 0.433        | 1.17E-04       | *0.434*       | 34.60         | 49.33         | 860.25        |
> | **with complete EPI**    | **13.60**     | **0.452**    | **1.02E-04**   | **0.430**     | **26.11**     | **32.23**     | **703.87**    |
>
> **Table II:** Quantitative results of the ablation study. The best and second-best results for each column are **bold** and *italicized*, respectively.
>
> ***
>
> **Comment 4: The effectiveness of the Implicit Pose Indicator (IPI) is also in question. The motivation for the IPI is that sparse keypoints lack image-level details, while IPI aims to retrieve richer information. However, Table 7 and 8 indicate that Animate-X achieves comparable performance to Animate-Anyone and UniAnimate on human videos. This suggests that the IPI does not provide any benefits for human animation.**
>
> The effectiveness of the Implicit Pose Indicator (IPI) have been demonstrated through the quantitative results in **Table III** (*i.e.*, Table 4 in the original submission) and the qualitative analysis in Figure 7 in the original submission.
>
> | **Method**        | **PSNR*** ↑   | **SSIM** ↑     | **L1** ↓         | **LPIPS** ↓     | **FID** ↓       | **FID-VID** ↓  | **FVD** ↓       |
> |--------------------|---------------|----------------|------------------|-----------------|-----------------|----------------|-----------------|
> | w/o IPI           | 13.30         | 0.433          | 1.35E-04         | *0.454*         | 32.56           | 64.31          | 893.31          |
> | w/o LQ            | *13.48*       | 0.445          | 1.76E-04         | *0.454*         | 28.24           | 42.74          | 754.37          |
> | w/o DQ            | 13.39         | 0.445          | **1.01E-04**     | 0.456           | 30.33           | 62.34          | 913.33          |
> | **Animate-X**      | **13.60**     | **0.452**      | *1.02E-04*       | **0.430**       | **26.11**       | **32.23**      | **703.87**      |
>
> **Table III:** Quantitative results of the ablation study on IPI. The best and second-best results for each column are **bold** and *italicized*, respectively.
>
> **1)** The primary purpose of Animate-X is to animate universal characters, especially anthropomorphic figures in cartoons and games. Human animation is **NOT** the primary focus of this work as it is a small subset of 'X'. Table 7 & 8 verify that even for human figures, Animate-X's performance is on par with the latest works focusing on animating human figures. This strongly indicates the generalization capability of Animate-X.
>
> **2)** IPI does retrieve richer information from driven video that is critical to some hard cases that lack of enough details in anthropomorphic figures, e.g., . It is reasonable that its contribution is marginal for those simple human-driven animations that the details are already sufficient to capture human motion, which are not the cases that IPI is designed to address. Therefore, for datasets like TikTok with exclusive human data only, we just want to show II also improves a bit and Animate-X is well backward compatible for human figures;
>
> **3)** Anthropomorphic characters are arguably more desirable in gaming film and short videos. Therefore we introduce a novel benchmark beyond human, as detailed in Section 3.4. We kindly suggest the reviewer to watch the MP4 videos in the updated supplementary materials.

---

> ### Author Response · Authors · 2024-11-25
>
> Dear Reviewer aHUH,
>
> Thank you again for the great efforts and valuable comments. We hope you find the response satisfactory. As the discussion phase is about to close, we are eagerly looking forward to hearing from you regarding any further feedback. We will be more than happy to address any additional concerns you may have.
>
> Best,
>
> Animate-X Authors

---

> ### Comment · Reviewer_aHUH · 2024-11-25
>
> Thanks for your feedbacks which address most of my concerns. However, I am still disagree with that T2I + I2V is the optimal way to constitute the benchmark for character animation task. Utilizing this approach for training data is acceptable; however, it is not particularly appropriate for benchmarking purposes as we have more higher standard, that is, the ground-truth, for benchmark samples. The authors give many quantitative values to demonstrate the feasibleness, but the values sometimes fail to align with human preference. Specifically, the provided videos in the benchmark show some undesirable artifacts (blurring, twisting, etc.) around the hands and feet, and tend to show similar motion patterns without complex scenarios due to the limitation of current I2V models. I think a better way of creating a character benchmark is to create 3D models and render them with predefined actions with 3D tools such as Blender, Maya. Of course, it is more laborious, expensive and requires expert skills.
>
> Anyway, I will raise my score to 8 based on the explanation from the authors for the good performance, the writing and the design of a pose pool.

---

> > ### Author Response · Authors · 2024-11-25
> >
> > Thank you for taking the time to review our revisions and for your willingness to raise the score to 8.
> >
> > We agree that creating 3D models and rendering them with predefined actions using tools like Blender and Maya is a superior approach for developing a character benchmark. In fact, we are currently making preparations to utilize 3D models to produce animated videos that will showcase a wider array of motion patterns and more complex scenarios to support our benchmark.
> >
> > Once again, we appreciate your suggestions and your acknowledgment of the authors' explanations regarding the strong performance, writing quality, and the design of the pose pool.

---

> > ### Author Response · Authors · 2024-11-29
> > **Looking forward to your improving the final score in the system**
> >
> > Dear Reviewer aHUH,
> >
> > Thank you for your efforts and valuable comments on improving the quality of our manuscript. We kindly remind that the discussion time is coming to an end. We are also glad to hear that you are willing to raise the score, and we look forward to your editing the final score in the system. Thank you again for your efforts and please feel free to contact us if you have any further questions.
> >
> > Best regards,
> >
> > The  Authors of Animate-X

---

### Author Response · Authors · 2024-11-20
**We sincerely thank all reviewers for their careful reading and constructive comments.**

We sincerely thank all reviewers for their careful reading and constructive comments, which have been invaluable in improving our work. We also deeply appreciate the reviewers’ acknowledgment of:
- The notable improvements across various characters compared to existing animation methods (aHUH, feUz, ESK8)
- Comprehensive experiments and ablation studies (aHUH and mbHE)
- The introduction of A2Bench (mbHE, feUz, ESK8)
- The proposed novel module (feUz, ESK8)

In response to the reviewers' comments, we have re-uploaded our supplementary materials, which include the **complete responses** (at **.zip/Animate-X_rebuttal_response_letter.pdf**) along with the relevant figures and tables. The response letter is also contained in the **main paper, after page 25**. We sincerely invite the reviewers to refer to these materials for a better reading experience. We hope that our response satisfactorily addresses your concerns.

---

### Meta-Review · Area_Chair_1r4d · 2024-12-22

**Metareview:**

The paper received very positive ratings from the reviewers. They highlight improved results, new ideas, the introduction of a new benchmarks, as well as the ability of the method to animate objects not having a distinct skeleton structure. The reviewers also pointed out that at times the paper can be hard to follow, or can be lacking the necessary analysis. This led to a lengthy discussion between the authors and reviewers during which most of concerns were addressed, leading to improved scores. The AC agrees, the manuscript presents an interesting piece of work. Congrats!

**Additional Comments On Reviewer Discussion:**

There was a very healthy discussion between the reviewers and the authors. New ablations were shown, unclear parts were explained, certain analyses were given.

---

### Decision · Program_Chairs · 2025-01-22

Accept (Poster)